# The lipoprotein-associated phospholipase A2 inhibitor Darapladib sensitises cancer cells to ferroptosis by remodelling lipid metabolism

Mihee Oh[1,17], Seo Young Jang [2,17], Ji-Yoon Lee [3,17], Jong Woo Kim[3,4], Youngae Jung[2], Jiwoo Kim[5,6], Jinho Seo[7], Tae-Su Han[8], Eunji Jang[9], Hye Young Son [10,11], Dain Kim[2,12], Min Wook Kim[3], Jin-Sung Park[13], Kwon-Ho Song[14], Kyoung-Jin Oh [3,4], Won Kon Kim [3,4], Kwang-Hee Bae [3,4], Yong-Min Huh[9,10,11], Soon Ha Kim[13], Doyoun Kim [5], Baek-Soo Han[1,3,4 ✉], Sang Chul Lee [3 ✉], Geum-Sook Hwang [2,15 ✉] & Eun-Woo Lee [3,4,16 ✉]

Arachidonic and adrenic acids in the membrane play key roles in ferroptosis. Here, we reveal that lipoprotein-associated phospholipase A2 (Lp-PLA2) controls intracellular phospholipid metabolism and contributes to ferroptosis resistance. A metabolic drug screen reveals that darapladib, an inhibitor of Lp-PLA2, synergistically induces ferroptosis in the presence of GPX4 inhibitors. We show that darapladib is able to enhance ferroptosis under lipoprotein-deficient or serum-free conditions. Furthermore, we find that Lp-PLA2 is located in the membrane and cytoplasm and suppresses ferroptosis, suggesting a critical role for intracellular Lp-PLA2. Lipidomic analyses show that darapladib treatment or deletion of *PLA2G7*, which encodes Lp-PLA2, generally enriches phosphatidylethanolamine species and reduces lysophosphatidylethanolamine species. Moreover, combination treatment of darapladib with the GPX4 inhibitor PACMA31 efficiently inhibits tumour growth in a xenograft model. Our study suggests that inhibition of Lp-PLA2 is a potential therapeutic strategy to enhance ferroptosis in cancer treatment.

Ferroptosis is a newly identified form of programmed necrosis that requires free active iron[1,2]. Excessive accumulation of reactive oxygen species (ROS) in membrane phospholipids (PLs) is a main cause of ferroptosis; thus, lipid ROS are a distinct hallmark of ferroptosis[3]. Lipid ROS are primarily controlled by glutathione peroxidase 4 (GPX4), an enzyme that directly reduces lipid peroxides in membrane phospholipids to lipid alcohols[4,5]. GSH is simultaneously oxidised by GPX4 and thus acts as an essential cofactor of GPX4[6]. Intracellular levels of GSH, which consists of cysteine, glutamate, and glycine, are mainly controlled by the cystine–glutamate antiporter. Based on this, several ferroptosis-inducing agents (FINs) that target GPX4 or cystine-glutamate antiporters have been developed and are regarded as potential drugs for cancer treatment[2,7]. While the GPX4/GSH system protects most types of cells from ferroptosis, recent studies suggest that chemoresistant cancer cells with mesenchymal characteristics or drug-tolerant cancer cells are particularly susceptible to ferroptosis[8–11]. Therefore, ferroptosis has been widely recognised as an emerging anticancer treatment strategy. In addition, ferroptosis has been implicated in various human diseases, such as ischaemia–reperfusion injury in the kidneys, heart, brain, and liver; thus, ferroptosis inhibitors have been developed for the treatment of these diseases[12].

As ferroptosis is induced by lipid peroxidation, various metabolic and signalling pathways that affect lipid peroxidation, such as lipid metabolism, iron metabolism, and antioxidant pathways, are involved in the ferroptosis pathway[6,12–15]. In particular, arachidonic acid (AA; C20:4) and adrenic acid (AdA; C22:4), which are linked in membrane phospholipids, especially phosphatidylethanolamine (PE) and phosphatidylcholine (PC), are the most critical lipids for lipid peroxidation and ferroptosis[16,17]. In this regard, the critical roles of long-chain acyl-CoA synthetase 4 (ACSL4) and lysophospholipid acyltransferase (LPCAT3), which mediate the incorporation of AA into PE or PC, in ferroptosis in various pathological contexts have been the focus of considerable research[2,6,18]. AA can be directly imported from extracellular fluid supplied from the blood vessels, and then further elongated into AdA by ELOVL5[3,6]. In addition, AA can be synthesised from the n-6 essential fatty acid linoleic acid (LA, C18:2), the most abundant fatty acid in serum and plasma. Our recent study suggests that certain gastric cancer cells depend on this PUFA synthesis pathway and thus show hypersensitivity to ferroptosis[8]. Although intracellular AA levels are important for ferroptosis, only a small fraction of AA in phospholipids is oxidised under ferroptotic conditions[16,17]. However, little is known about how cells maintain phospholipids containing AA to regulate lipid peroxidation and ferroptosis.

Phospholipase A2s (PLA2s) are enzymes that hydrolyse fatty acids at the sn-2 position of phospholipids, and there are more than 50 enzymes in the PLA2 superfamily[19]. PLAs are largely classified into secretory PLA2 (sPLA2), cytosolic PLA (cPLA), $Ca^{2+}$-independent PLA2 (iPLA2), and lipoprotein-associated phospholipase A2 (Lp-PLA2; encoded by PLA2G7 and known as plasma platelet-activating factor acetylhydrolase [PAF-AH]), all of which have different sequences and structures[20,21]. The major role of PLA2 is to extract AA from membrane phospholipids, thereby contributing to the formation of pathophysiological lipid mediators such as prostaglandins, leukotrienes, and lysophospholipids. Since AAs play a key role in ferroptosis, PLA2 is thought to be involved in ferroptosis. While cPLA2 is already known to increase lipid peroxide by releasing AA, its contribution to ferroptosis is unclear[22,23]. iPLA2β, which is encoded by PLA2G6, was recently identified to specifically cleave oxidised PE containing AA (15-HpETE-PE), which results in the suppression of ferroptosis[24–26].

In this study, we discovered that darapladib, an Lp-PLA2 inhibitor, sensitises cells to ferroptosis and shows antitumour activity when administered with PACMA31, a GPX4 inhibitor, in vivo. Using lipidomic analysis, we found that darapladib rewired lipid metabolism to render cells vulnerable to ferroptosis. Interestingly, intracellular Lp-PLA2, but not extracellular Lp-PLA2, seems to protect cells from ferroptosis. Thus, these results suggest that Lp-PLA2 is an essential negative regulator of ferroptosis sensitivity.

## Results

### A metabolic library screen identified darapladib, an Lp-PLA2 inhibitor, as a ferroptosis-sensitising drug

To identify the metabolic pathways regulating ferroptosis and discover metabolic drugs that kill cancer cells synergistically with ferroptosis inducers, 403 metabolism-modulating compounds were screened from libraries on the basis of their ability to modulate RSL3-induced ferroptosis (Fig. 1a). In this screen, we used Hs746T cells, mesenchymal-type gastric cancer cells that are refractory to standard therapy but are sensitive to ferroptosis, as shown in our previous study[8]. When used alone, several compounds reduced cell viability to less than 50%, but most compounds did not significantly alter the overall survival rate (Supplementary Fig. 1a). We focused on several compounds that have no toxicity when used alone but significantly promote cell death when administered with RSL3 and selected several candidates that acted synergistically with RSL3 (Supplementary Fig. 1a). After validation with ferrostatin-1 (Fer-1), a ferroptosis inhibitor, and investigation of the literature regarding known targets,

mechanisms, in vivo use, and clinical trials, darapladib (SB-480848), an Lp-PLA2 inhibitor, was chosen as the final candidate (Supplementary Fig. 1b). Because darapladib itself is toxic to cancer cells when used at high concentrations, 2 μM darapladib was used in the following experiment (Supplementary Fig. 2a).

Darapladib sensitised Hs746T and SNU-484 cells, both of which are mesenchymal-type gastric cancer cells, to ferroptosis (Fig. 1b–d)[8]. Notably, when cells were cotreated with darapladib, the half-maximal inhibitory concentration (IC50) values for RSL3 were drastically lowered in both cell lines (Fig. 1b). The marked decrease in cell viability observed after treatment with RSL3 combined with darapladib was almost completely eliminated by treatment with Fer-1, a ferroptosis inhibitor (Fig. 1b, c and Supplementary Fig. 2b). However, the pan-caspase inhibitor zVAD-fmk or the RIPK1 inhibitor necrostatin-1 (Nec-1) failed to rescue the viability of the cells (Supplementary Fig. 2b). Through PI uptake and LDH level assays, we confirmed that the decrease in cell viability was caused by necrotic cell death (Fig. 1e and Supplementary Fig. 2c, d). In addition, RSL3 induced cell death more efficiently when cells were plated at a low density, as previously reported (Fig. 1f)[27]. Under this condition, cell death was further enhanced by darapladib (Fig. 1f and Supplementary Fig. 2e). Furthermore, a darapladib concentration of 0.5 μM was sufficient to enhance GPX4 inhibitor-induced ferroptosis (Fig. 1g). These data suggest that darapladib specifically promotes ferroptosis induced by GPX4 inhibitors.

### Darapladib is a general activator of ferroptosis in various cancer cell types

When ferroptosis was induced with other GPX4 inhibitors, such as ML210 and JKE1674, a derivative of ML210, darapladib also enhanced ferroptosis in Hs746T and SNU-484 cells (Fig. 2a). Notably, the increase in cell death caused by the combined treatment was fully reversed by Fer-1, confirming that ferroptosis was indeed augmented by darapladib (Fig. 2b). In addition to gastric cancer cell lines, consistent synergistic effects of RSL3 and darapladib were observed in lung cancer cell lines such as A549 and H1299 and in the liver cancer cell line HepG2 (Supplementary Fig. 3a, b). We also found that ferroptotic cell death was promoted when RSL3 and darapladib were coadministered to YCC-16 cells, another type of anticancer treatment–refractory cell (Supplementary Fig. 3c, d). Furthermore, darapladib renders non-cancer cells, such as H9c2 and MEFs, susceptible to RSL3-induced ferroptosis, suggesting the conserved effect of darapladib on ferroptosis (Supplementary Fig. 3e).

We then questioned whether darapladib also sensitises cells to ferroptosis upon GPX4 depletion. Therefore, we established H1299 cell lines stably expressing lentiviral shRNA for GPX4, which was selected in the presence of Fer-1 to prevent ferroptosis upon GPX4 depletion (Supplementary Fig. 4a). The removal of ferrostatin-1 can induce lipid peroxidation and ferroptotic cell death, and these effects were reinforced by darapladib, confirming that darapladib sensitises against GPX4 depletion-induced ferroptosis (Supplementary Fig. 4b–e).

We then investigated whether darapladib also responds to other ferroptotic stimuli, such as GSH depletion. Culture in cysteine-depleted medium for 18 h did not greatly reduce cell viability, but culture in cysteine-depleted medium in the presence of darapladib caused rapid cell death in both Hs746T and SNU-484 cells (Fig. 2c). Again, cell viability was remarkably recovered by Fer-1, suggesting that darapladib also sensitises cells to cysteine deprivation-induced ferroptosis (Fig. 2c). In addition, darapladib promoted ferroptosis induced by erastin, an inhibitor of system $x_c^-$ (Supplementary Fig. 4f).

Subsequently, we measured the level of lipid peroxidation, a hallmark of ferroptosis, using C11 BODIPY[581/591]. The treatment of cells with 0.1 μM RSL3 induced no increase in lipid peroxidation compared with the level found in the control group. However, more than 80% of cells cotreated with darapladib and RSL3 showed oxidised

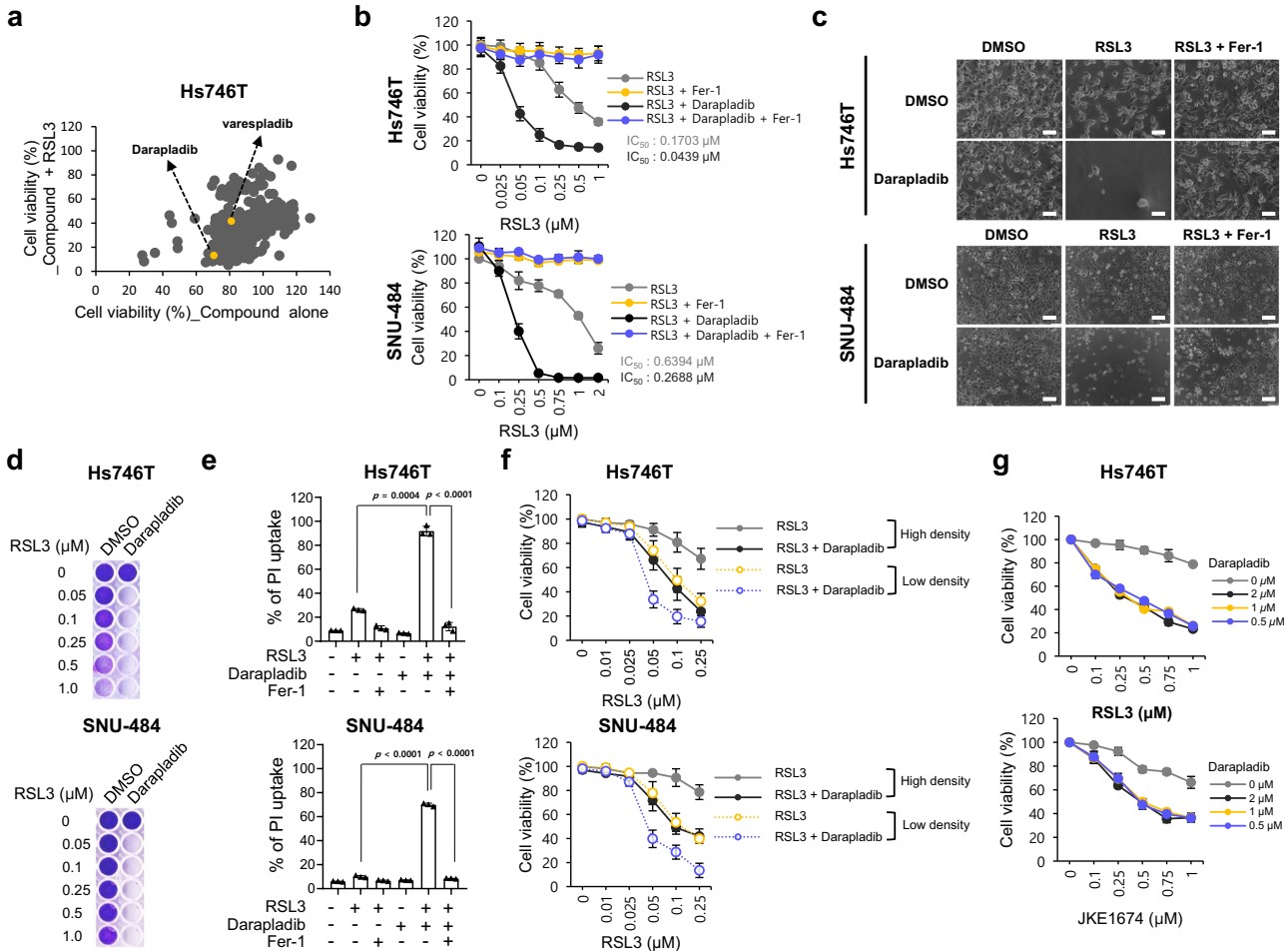

**Fig. 1 | Identification of darapladib as a ferroptosis-targeting drug by metabolic library screening. a** Relative viability of Hs746T cells treated with 10 µM compound alone or compound and 0.5 µM RSL3 for 24 h. **b** Relative viability of Hs746T and SNU-484 cells treated with increasing concentrations of RSL3 and/or 2 µM darapladib for 20 h. Cells were plated at 30,000 Hs746T cells/well and 40,000 SNU-484 cells/well in 200 µl of media. The data are presented as the means ± SDs (n = 6 independent experiments). **c** Images of cells treated with 0.2 µM RSL3 and/or 2 µM darapladib in the presence or absence of 1 µM Fer-1. Experiments were repeated three times. Scale bar, 200 µm. **d** Crystal violet staining of cells treated with RSL3 and/or 2 µM darapladib for 48 h. Cells were plated at 20,000 Hs746T cells/well and 25,000 SNU-484 cells/well in 200 µl of media. **e** PI uptake from Hs746T and SNU-484 cells treated with RSL3 and/or 2 µM darapladib in the presence or absence of 1 µM Fer-1 for 20 h. The data are presented as the means ± SDs (n = 3 independent experiments, the significance of the results was assessed using a two-tailed Student's t test). **f** Relative viability of cells at high (30,000 Hs746T cells/well and 40,000 SNU-484 cells/well) and low (20,000 Hs746T cells/well and 25,000 SNU-484 cells/well) densities upon RSL3 and 2 µM darapladib treatment. The data are presented as the means ± SDs (Hs746T: n = 3, SNU-484: n = 4 independent experiments). **g** Relative viability of Hs746T cells treated with RSL3 or JKE1674 in the presence of increasing concentrations of darapladib. The data are presented as the means ± SDs (n = 3 independent experiments). Exact p values provided as source data. Source data are provided as a source data file.

C11-BODIPY, as evidenced by flow cytometry and fluorescence microscopy (Fig. 2d, e and Supplementary Fig. 4g). These signals were completely abolished by Fer-1, confirming that lipid peroxidation was indeed enhanced. In addition, although oxidised C11 BODIPY signals transiently appeared before cells died at -14 h after cysteine deprivation, darapladib facilitated lipid peroxidation, as revealed by the observation of oxidised C11 BODIPY signals at -12 h (Fig. 2f). These data suggest that darapladib renders cells sensitive to ferroptosis by increasing lipid peroxidation.

### Darapladib sensitises cells to ferroptosis independent of lipoprotein

We next investigated whether darapladib sensitises cells to ferroptosis in an Lp-PLA2-dependent manner. Since Lp-PLA2 is known to be associated with lipoprotein[28], we first tested the effect of darapladib in lipoprotein-deficient conditions. When cells were cultured in lipoprotein-deficient human serum (LPDS), RSL3-induced ferroptosis was markedly diminished, and darapladib had no further effect when administered for 6 h (Fig. 3a, b). Although starvation stress may

activate the mTOR pathway, which can suppress ferroptosis, the levels of phospho-S6K were unaffected by lipoprotein deficiency (Supplementary Fig. 5a)[29,30]. Interestingly, supplementation with HDL, but not LDL or VLDL, resensitised cells to ferroptosis under lipoprotein deficiency, suggesting that HDL may contribute to ferroptosis, although the underlying mechanism is unclear (Supplementary Fig. 5b). Notably, treatment with RSL3 induced the death of cells cultured in LDPS with darapladib for 24 h but did not kill the control cells (Fig. 3c). These data imply that although lipoprotein deficiency generally slows the ferroptosis response, darapladib is still able to increase ferroptosis in the absence of lipoprotein. In addition, darapladib sensitised cells to ferroptosis under serum starvation conditions, supporting the indispensable role of lipoprotein in darapladib-mediated ferroptosis sensitisation (Fig. 3d). Interestingly, unlike lipoprotein deficiency, serum starvation did not alleviate ferroptosis, probably due to the concomitant depletion of anti-ferroptotic components in serum, such as vitamin E, selenium, and CoQ10, or a difference in composition between human and bovine serum.

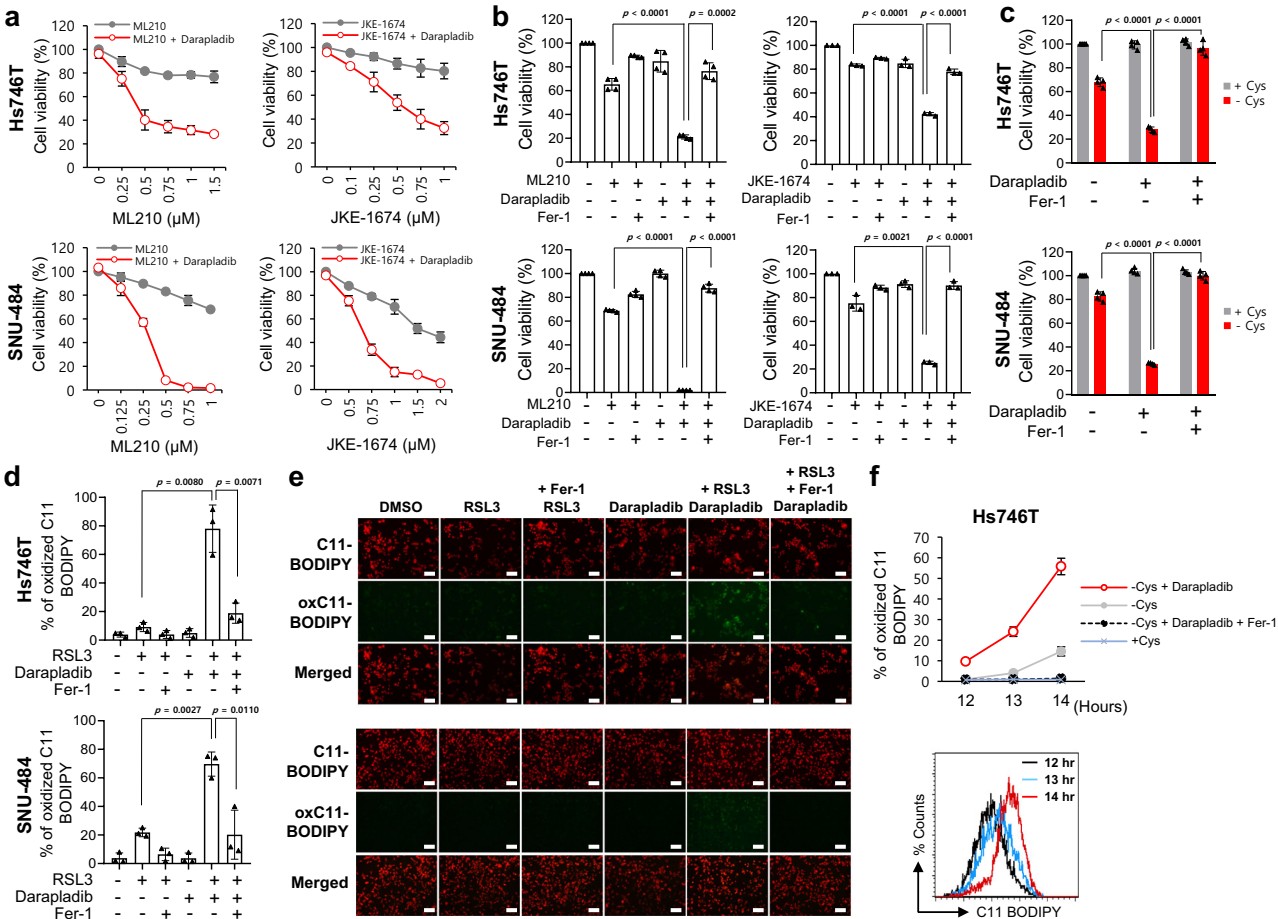

**Fig. 2 | Darapladib sensitises cells to ferroptosis induced by GPX4 inhibition or cysteine deprivation. a** Relative viability of Hs746T and SNU-484 cells treated with increasing concentrations of ML210 or JKE1674 in the presence or absence of 2 μM darapladib for 20 h. The data are presented as the means ± SDs ($n = 3$ or 4 independent experiments). **b** Relative viability of cells in the presence or absence of 1 μM Fer-1. The data are presented as the means ± SDs ($n = 3$ or 4 independent experiments with, the significance of the results was assessed using a two-tailed Student's $t$ test). **c** Viability of Hs746T and SNU-484 cells cultured in cysteine-deficient medium in the presence and absence of 2 μM darapladib for 18 h. The data are presented as the means ± SDs ($n = 4$ independent experiments with, the significance of the results was assessed using a two-tailed Student's $t$ test). **d**, **e** Lipid peroxidation levels in Hs746T and SNU-484 cells treated with 0.2 μM RSL3 and 2 μM darapladib for 1 h. Lipid oxidative potential was assessed by flow cytometry (**d**) and microscopy (**e**) using C11-BODIPY[581/591]. Probes fluorescing green represent those that have been oxidised. Experiments were repeated three times. Scale bar, 200 μm. **d** The data are presented as the means ± SDs ($n = 3$ independent experiments with, the significance of the results was assessed using a two-tailed Student's $t$ test). **f** Lipid peroxidation levels in Hs746T cells cultured in cysteine-deficient medium in the presence or absence of darapladib for 12–14 h. The data are presented as the means ± SDs ($n = 3$ independent experiments). Exact $p$ values provided as source data. Source data are provided as a source data file.

## Darapladib does not affect known ferroptosis regulators or iron levels

We then explored the mechanism by which darapladib sensitises cells to ferroptosis. First, we investigated the expression of well-known ferroptosis regulators, such as GPX4, NRF2, FSP1, ACSL4, LPCAT3, PEBP1, FADS1, and ELOVL5[6,31]. As previously reported, GPX4 bands were shifted due to covalent binding to RSL3[32], but the levels of GPX4 were unchanged (Fig. 3e). In addition, we observed no significant differences in protein expression upon RSL3 and/or darapladib treatment (Fig. 3e). Furthermore, darapladib does not affect the expression levels of NRF2 target genes, which is crucial for ferroptosis suppression (Supplementary Fig. 6a)[33–35]. Since the availability of free iron is also a key factor for ferroptosis execution[36], we also investigated the free iron levels upon darapladib treatment. However, darapladib did not regulate intracellular iron levels (Fig. 3f).

Lp-PLA2 is primarily produced by monocytes/macrophages and catalyses the hydrolysis of oxidised phospholipids (oxPLs), mainly PC on LDL, into lysophosphatidylcholine (LysoPC) and oxidised nonesterified fatty acids (oxNEFAs)[37]. In particular, lysoPC induces ROS production, contributing to atherosclerosis and plaque destabilisation[38–40]. Therefore, Lp-PLA2 is regarded as a risk factor for cardiovascular disease, and darapladib (SB-480848) was developed to specifically inhibit Lp-PLA2 to treat atherosclerosis, although darapladib recently failed in phase III clinical trials due to lack of efficacy[41–43]. As lysoPC has previously been shown to promote AA release from endothelial cells[44], we hypothesised that darapladib reduces the extracellular levels of lysoPC, thereby abolishing the export of AA and eventually facilitating ferroptosis. Unexpectedly, however, the levels of lysoPC and AA in the medium were slightly increased or remained unchanged upon darapladib treatment (Supplementary Data 5, Supplementary Fig. 6b). Interestingly, medium supplemented with FBS contained high levels of lysoPCs, which decreased rapidly over time during incubation and were probably absorbed into cells and used for phospholipid synthesis through the Lands cycle[45]. Although we cannot rule out the possibility that Lp-PLA2-mediated lysoPC production remains linked to ferroptosis, our findings point to the existence of another dominant mechanism through which Lp-PLA2 regulates ferroptosis.

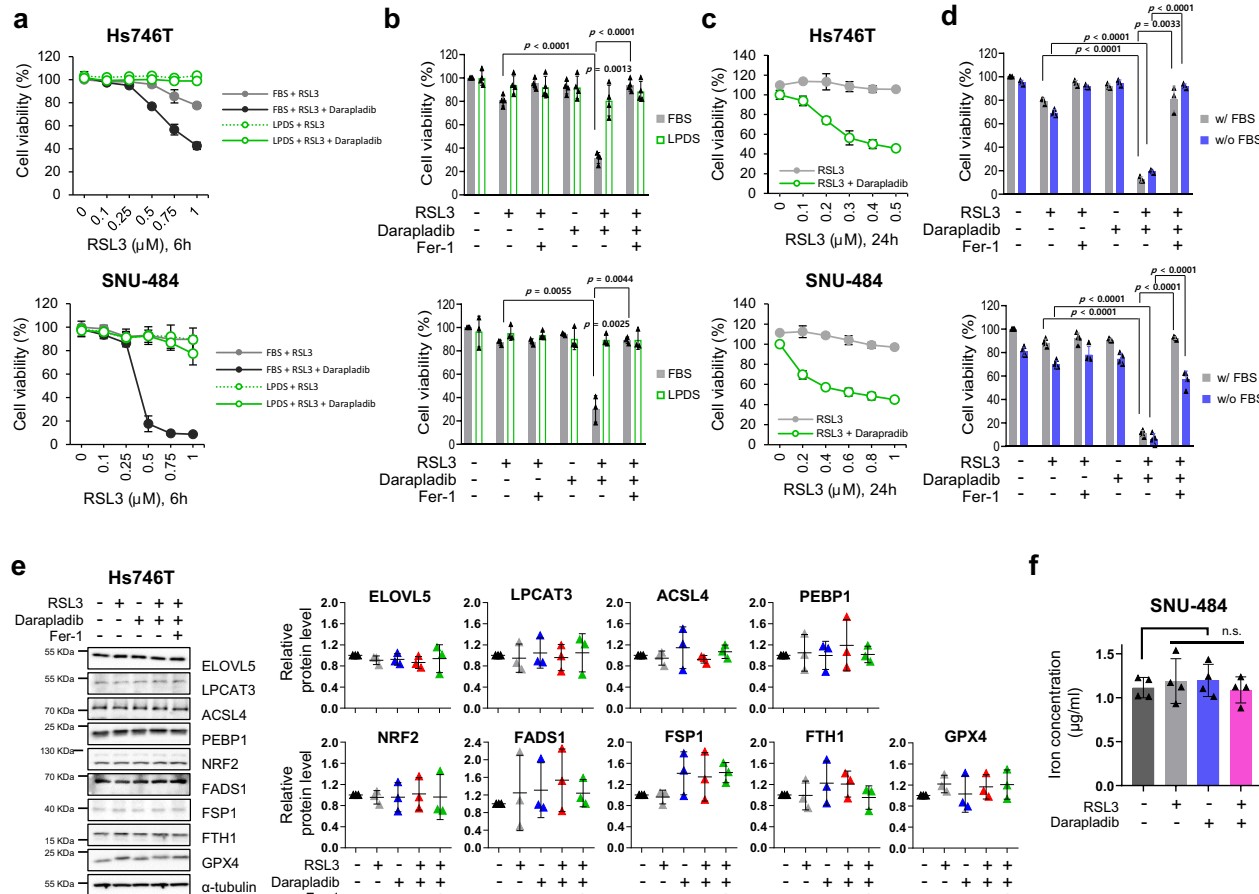

**Fig. 3 | Darapladib is able to promote ferroptosis in the absence of lipoprotein. a**, **b** Relative viability of Hs746T and SNU-484 cells treated with RSL3 and 2 μM darapladib cultured in medium containing FBS or LPDS for 6 h in the presence or absence of Fer-1. The data are presented as the means ± SDs ($n = 3$ or 4 independent experiments with, the significance of the results was assessed using a two-tailed Student's $t$ test). **c** Relative viability of cells in the indicated medium for 24 h. The data are presented as the means ± SDs ($n = 3$ independent experiments). **d** Relative viability of cells treated with RSL3 and 2 μM darapladib in the presence or absence of FBS. The data are presented as the means ± SDs ($n = 3$ or 4 independent experiments with, the significance of the results was assessed using a two-tailed Student's

$t$ test). **e** Western blots showing the expression levels of well-known ferroptosis regulators upon 0.2 μM RSL3 and 2 μM darapladib treatment as indicated. Three blots were cut according to protein size and directed to western blotting, and reprobed to detect proteins of similar size. Each protein was normalised to α-tubulin on the same blot. Experiments were repeated three times and the data are also presented as the means ± SDs ($n = 3$ independent experiments). **f** Total iron level measured in the lysates of cells treated with 0.2 μM RSL3 and 2 μM darapladib. The data are presented as the means ± SDs ($n = 4$ independent experiments, with n.s. nonsignificant compared to the control with two-tailed Student's $t$ test). Exact $p$ values provided as source data. Source data are provided as a source data file.

## Lp-PLA2 inhibits ferroptosis

Because lipoproteins and extracellular lysoPCs are dispensable for darapladib-induced ferroptosis, we wondered whether Lp-PLA2 is indeed associated with the ferroptosis pathway. Therefore, we employed small interfering RNA (siRNA) pools containing four different siRNAs against *PLA2G7*, which encodes Lp-PLA2, and efficient knockdown by the siRNA was confirmed at the mRNA level (Supplementary Fig. 7a). Consistent with darapladib, depletion of Lp-PLA2 also promoted RSL3-induced ferroptosis in both Hs746T and SNU-484 cells (Supplementary Fig. 7b).

We subsequently established H1299 and YCC-16 cells with *PLA2G7* deficiency using CRISPR–Cas9 systems and confirmed complete knockout at the mRNA level (Fig. 4a and Supplementary Fig. 8a). Again, *PLA2G7* KO cells were more sensitive to RSL3 than parental cells, and the increased sensitivity to ferroptosis was reversed by Fer-1 (Fig. 4b–g and Supplementary Fig. 8b–f). In addition, the level of lipid peroxidation was increased in *PLA2G7* KO cells (Fig. 4h and Supplementary Fig. 8g). Furthermore, under cysteine-deficient conditions, wild-type cell viability was not greatly reduced, but *PLA2G7* KO cells underwent rapid cell death, which was attenuated by Fer-1 (Fig. 4i, j and Supplementary Fig. 8h). Similar to the findings with darapladib, *PLA2G7*-

deleted cells showed no significant alterations in several key ferroptosis regulators although some fluctuations in the protein levels were observed (Supplementary Fig. 8i). Moreover, *PLA2G7* KO cells exhibited reduced sensitivity to darapladib-enhanced ferroptosis, supporting the requirement of Lp-PLA2 for darapladib (Fig. 4k and Supplementary Fig. 8j). These data imply that Lp-PLA2 produced in cells rather than in serum might play a key protective role in ferroptosis.

We next explored whether ectopic expression of Lp-PLA2 protein in cells could ameliorate RSL3-induced ferroptosis. Hs746T cells overexpressing Lp-PLA2 exhibited reduced susceptibility to RSL3-induced cell death (Fig. 4l and Supplementary Fig. 7k). In addition, overexpression of Lp-PLA2 reduced RSL3-induced lipid peroxidation (Fig. 4m and Supplementary Fig. 7l). Taken together, these findings suggest that Lp-PLA2 is a bona fide negative regulator of ferroptosis.

## Intracellular Lp-PLA2 is responsible for ferroptosis sensitisation

Although Lp-PLA2 is known to be secreted into the extracellular matrix and to interact with lipoprotein[46], our data suggest that lipoprotein is not essential for darapladib-induced sensitisation to ferroptosis. In addition, several previous reports have implied that Lp-PLA2 is

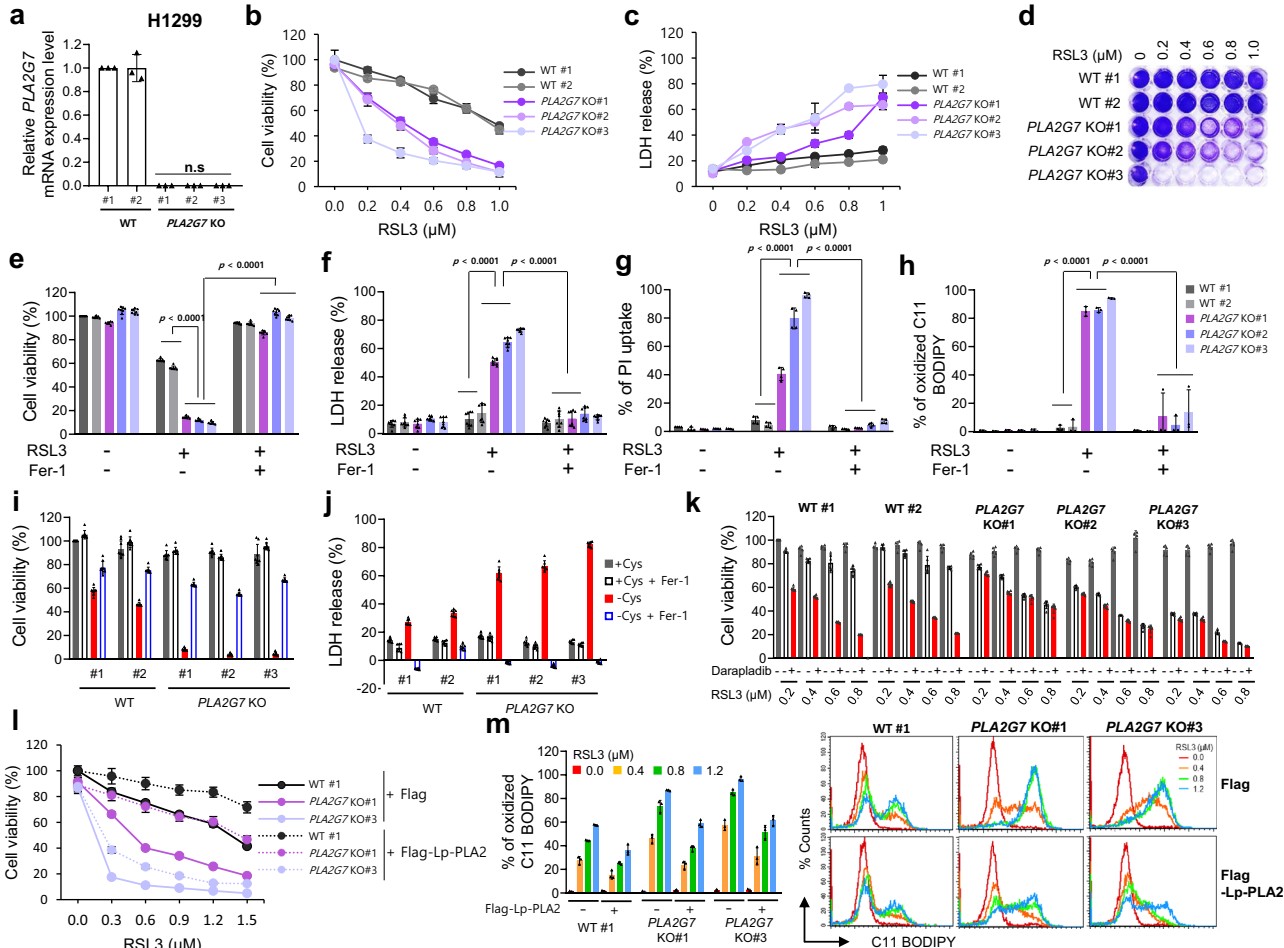

**Fig. 4 | Lp-PLA2 negatively regulates lipid peroxidation and ferroptosis.**
**a** Relative mRNA expression levels in WT and *PLA2G7* KO H1299 cells normalised to the β-actin expression levels. The data are presented as the means ± SDs (*n* = 3 independent experiments with, the significance of the results was assessed using one-sided Wilcoxon rank-sum test). **b, c** Relative cell viability and LDH levels of WT and *PLA2G7* KO H1299 cells treated with RSL3 for 20 h. The data are presented as the means ± SDs (**b**: *n* = 6, **c**: *n* = 4 independent experiments). **d** Crystal violet staining of WT and *PLA2G7*KO H1299 cells treated with various concentrations of RSL3. **e–g** Relative cell viability, LDH levels and PI uptake of WT and *PLA2G7* KO H1299 cells treated with RSL3 in the presence or absence of Fer-1 for 20 h. The data are presented as the means ± SDs (**e, f**: *n* = 8, **g**: *n* = 4, independent experiments with, the significance of the results was assessed using one-sided Wilcoxon rank-sum test). **h** Lipid peroxidation level of WT and *PLA2G7* KO H1299 cells treated with

RSL3 for 1.5 h. The data are presented as the means ± SDs (*n* = 3 independent experiments with, the significance of the results was assessed using one-sided Wilcoxon rank-sum test). **i, j** Cell viability and LDH level of WT and *PLA2G7* KO H1299 cells cultured in cysteine-deficient medium for 48 h. The data are presented as the means ± SDs (**i**: *n* = 8, **j**: *n* = 7 independent experiments). **k** Relative cell viability of WT and *PLA2G7* KO H1299 cells treated with various concentrations of RSL3 and/or 0.5 μM darapladib for 48 h. The data are presented as the means ± SDs (*n* = 6 independent experiments). **l, m** Relative viability and lipid peroxidation levels of WT and *PLA2G7* KO H1299 cells ectopically expressing Lp-PLA2 and treated with RSL3. The data are presented as the means ± SDs (**l**: *n* = 5, **m**: 3 independent experiments). Exact *p* values provided as source data. Source data are provided as a source data file.

localised in the cytoplasm[47–49]. We also observed a comparable amount of ectopic Lp-PLA2 in cell pellets compared to medium (Fig. 5a). In addition, overexpressed Lp-PLA2 existed in both the cytoplasm and plasma membrane (Fig. 5b). To exclude the involvement of extracellular Lp-PLA2, the medium of the cells was replaced with serum-free medium to remove the existing Lp-PLA2 in the medium, and the cells were immediately stimulated with RSL3. Interestingly, darapladib was still able to increase lipid peroxidation levels at 30 min after RSL3 treatment (Fig. 5c). Similarly, *PLA2G7* KO cells were more sensitive to ferroptosis than control cells in the presence of fresh medium without serum (Fig. 5d, e). These data suggest that intracellular Lp-PLA2 might have a protective role against ferroptosis.

### Deletion or inhibition of Lp-PLA2 induces lipid remodelling that is favourable to ferroptosis

Because intracellular PUFAs in phospholipids are crucial for ferroptosis sensitivity, we hypothesise that intracellular Lp-PLA2 might control

the lipid composition in cells. The assessment of lipidomic changes under *PLA2G7* deficiency by LC–MS revealed that the levels of PEs were generally increased in *PLA2G7* KO cells, whereas the lysophosphatidylethanolamine (lysoPE) and lysoPC levels were decreased in these cells (Fig. 5f, g and Supplementary Fig. 9a, b). A marked increase in the levels of PE species with three or more double bonds compared with those of PE species with fewer double bonds was observed (Fig. 5f). In particular, we also found that well-established pro-ferroptotic phospholipids such as PE-38:4 (18:0_20:4) were enriched in *PLA2G7* KO cells (Supplementary Fig. 9b).

We then tested whether the inhibition of Lp-PLA2 by darapladib results in the similar lipidomic changes observed in *PLA2G7* KO cells. A lipid profile analysis revealed that the levels of PEs were generally increased in darapladib-treated cells, whereas the free fatty acid (FFA) and lysoPE levels were decreased by darapladib (Fig. 6a and Supplementary Fig. 10a–c). These lipidomic changes were observed as early as 1 h and lasted for 4 h, indicating that phospholipid remodelling,

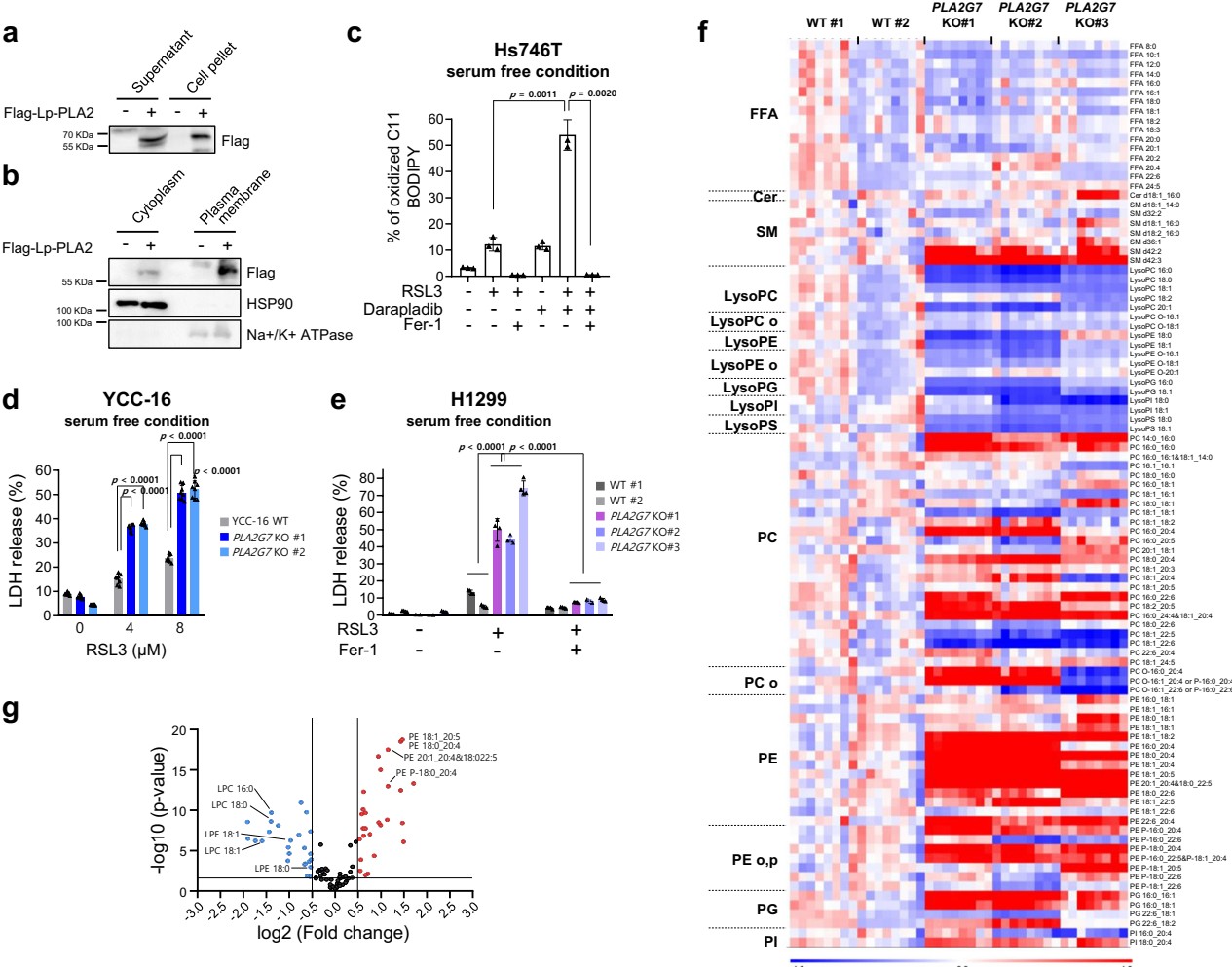

**Fig. 5 | Intracellular Lp-PLA2 is responsible for ferroptosis suppression by regulating phospholipid compositions. a** Western blot analysis to confirm the presence of Lp-PLA2 in the pellet and supernatant after ectopic expression of Lp-PLA2 in Hs746T cells. **b** Western blot analysis to confirm the presence of Lp-PLA2 in the cytoplasm and plasma membrane after ectopic expression of Lp-PLA2 in Hs746T cells. **c** Lipid peroxidation levels in Hs746T cells treated with RSL3 and/or 2 μM darapladib for 0.5 h and cultured in FBS-deficient medium. Lipid oxidative potential was assessed by flow cytometry and microscopy using C11-BODIPY[581/591]. The data are presented as the means ± SDs ($n = 3$ independent experiments with, the significance of the results was assessed using a two-tailed Student's $t$ test). **d, e** LDH levels of WT and *PLA2G7* KO cells treated with RSL3 in the absence of serum for 24 h. The data are presented as the means ± SDs (**d**: $n = 8$, **e**: $n = 4$ independent experiments with, the significance of the results was assessed using a two-

tailed Student's $t$ test (**d**) or one-sided Wilcoxon rank-sum test (**e**)). **f, g** Lipidomic analysis acquired from the UPLC/QTOF MS spectra of WT and *PLA2G7* KO H1299 cells. Each value in the heatmap is a coloured representation of a calculated Z score (**f**). Lipids are presented as the total numbers of carbon atoms, double bonds, and fatty acyl chains. $n = 8$ independent experiments. A volcano plot of lipid classes showing the log2(fold change) and −log10($p$) values in the control versus *PLA2G7* KO H1299 cells is presented (**f**). FFA free fatty acid, Cer ceramide, SM sphingomyelin, lysoPC lysophosphatidylcholine, lysoPE lysophosphatidylethanolamine, lysoPG lysophosphatidylglycerol, lysoPI lysophosphatidylinositol, lysoPS lysophosphatidylserine, PC phosphatidylcholine, PE phosphatidylethanolamine, PG phosphatidylglycerol, PI phosphatidylinositol, PS phosphatidylserine. Exact $p$ values provided as source data. Source data are provided as a source data file.

known as the Lands cycle, occurs very rapidly within the cell. In addition to its effect on diacyl PE species, darapladib induced the accumulation of ether lipids such as PE plasmalogen (PE-p) species, which are crucial for ferroptosis, and reduced the levels of lysoPE-p species[50,51] (Supplementary Fig. 10a–c). In addition, other phospholipids, such as PI, PS, and PG, may be the target of Lp-PLA2 because these PLs and lysoPLs are oppositely regulated by Lp-PLA2 deficiency or inhibition (Figs. 5, 6 and Supplementary Figs. 9, 10). In *PLA2G7* KO cells, PC species, the most abundant phospholipids, remained unchanged in cells treated with darapladib (Supplementary Fig. 10a–c). These data suggest that inhibition of Lp-PLA2 leads to the accumulation of AA and AdA containing PE or PE-p, rendering cells sensitive to ferroptosis although lipidomic changes under simultaneous inhibition of Lp-PLA2 and GPX4 were not directed determined.

Interestingly, the levels of saturated fatty acids and monounsaturated fatty acids (MUFAs), which are also generated by the de novo fatty acid synthesis pathway, were reduced by darapladib. Therefore, we evaluated the expression of several proteins, such as SREBP1, ACC1, and FASN. We confirmed that darapladib treatment reduced the precursor form of SREBP1 and significantly increased the phosphorylation of ACC, suggesting that the de novo fatty acid synthesis pathway was generally inhibited (Supplementary Fig. 10d). Notably, the reduction in oleic acid (OA, C18:1), which can strongly suppress ferroptosis[52,53], was markedly more pronounced than the reductions in saturated fatty acids, which may contribute to ferroptosis. These data imply that darapladib mediates the reduction in MUFA levels and accumulation of PE-PUFAs, thereby promoting ferroptosis.

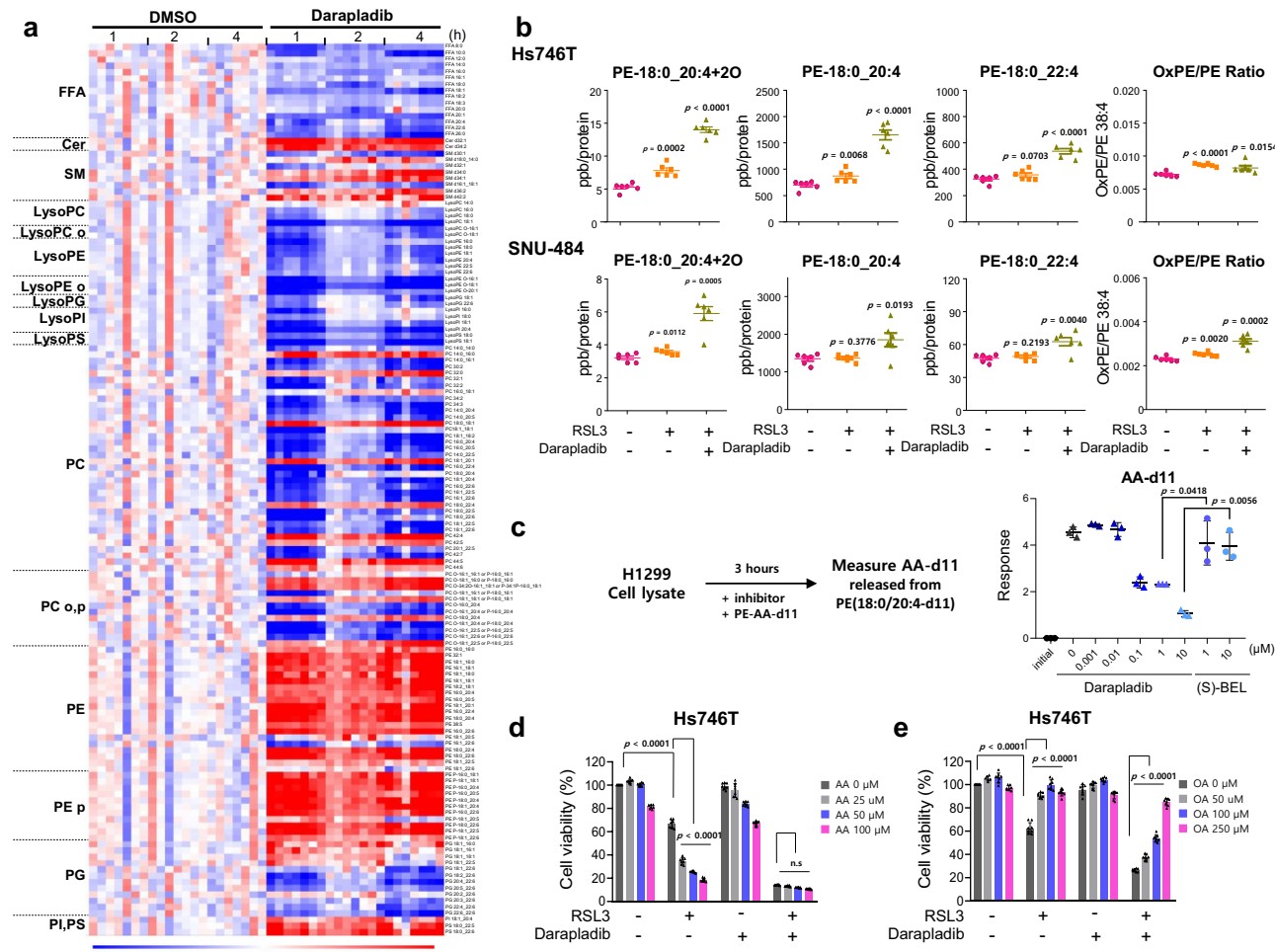

**Fig. 6 | Darapladib promotes ferroptosis by protecting against PE cleavage.**
**a** Lipidomic analysis based on the UPLC/QTOF MS spectra of Hs746T cells treated with 2 μM darapladib for 1, 2, and 4 h as described in Fig. 5. $n = 7$ independent experiments. **b** Concentrations of oxidised PE-18:0_20:4, PE-18:0_20:4, and PE-18:0_22:4 in Hs746T cells er treatment with RSL3 and/or 2 μM darapladib for 4 h and in SNU-484 cells after treatment with RSL3 and/or 2 μM darapladib for 3 h. The ratios of oxidised to non-oxidised PE-18:0_20:4 are also shown. The concentrations were determined by LC–MS/MS and were normalised to the cellular protein level. The data are presented as the means ± SDs ($n = 6$ independent experiments, the significance of the results was assessed using a two-tailed Student's $t$ test). **c** PE-

18:0_20:4-d11 cleavage analysis of cell lysates incubated with the indicated concentrations of darapladib and (S)-BEL for 3 h and subjected to detection of AA-d11 by LC–MS/MS. The data are presented as the means ± SDs ($n = 3$ independent experiments, the significance of the results was assessed using a two-tailed Student's $t$ test). **d, e** Relative viability of Hs746T cells pretreated with AA or OA for 20 h and treated with RSL3 and/or 2 μM darapladib for 20 h. The data are presented as the means ± SDs ($n = 8$ independent experiments, the significance of the results was assessed using a two-tailed Student's $t$ test). Exact $p$ values provided as source data. Source data are provided as a source data file.

Recently, Lp-PLA2 was shown to specifically cleave oxidised 1-stearoyl-2-arachidonyl-PE (SAPE; hereinafter referred to as PE-18:0_20:4) and thereby abrogated engulfment of ferroptotic cells by macrophages[54]. We therefore investigated whether darapladib also protects oxidised PE-18:0_20:4 (oxPE-18:0_20:4) from cleavage into PE and FFA by Lp-PLA2 upon ferroptosis induction. Although the levels of oxPE-18:0_20:4 were slightly increased in response to a low concentration of RSL3 or darapladib, cotreatment with RSL3 and darapladib markedly increased the levels of oxPE-18:0_20:4 in both Hs746T and SNU-484 cells (Supplementary Data 3, 6, Fig. 6b and Supplementary 11a). However, this induction was accompanied by increases in non-oxidised PE-18:0_20:4 and in PE-18:0_22:4, as supported by our targeted lipid analysis. Nevertheless, a slight increase in the ratio of oxPE-18:0_20:4 to PE-18:0_20:4 was also observed after darapladib treatment (Fig. 6b and Supplementary Fig. 11a), suggesting that Lp-PLA2 activity against oxidised SAPE also partially contributes to ferroptosis sensitivity. Consistently, *PLA2G7* KO cells exhibited a modest increase in their oxPE-18:0_20:4 levels, possibly due to the observed increase in PE-18:0_20:4, and RSL3 treatment significantly

elevated the oxPE-18:0_20:4 levels (Supplementary Data 7, Supplementary Fig. 11b).

Despite its known specificity towards PC lipids with oxidised short chain fatty acids in vitro[55], recent research suggests that Lp-PLA2 can also cleave oxidised forms of PS and PE[54,56]. These findings imply that Lp-PLA2 has a wide-ranging substrate specificity in vivo, which aligns with the results of our lipid profile analysis (Figs. 5 and 6). Our molecular docking and structural analysis revealed that the active pocket of Lp-PLA2 can accommodate various types of lipids (Supplementary Fig. 12a, b). Additionally, the rescored binding energy (DG) for oxPE, PC, PE, PG, PS, and PI were −5.16, −5.38, −5.26, −5.34, −5.08, and −5.34 kcal/mol, respectively (Supplementary Fig. 12c). The different types of lipid head groups are responsible for the notable differences in binding modes and the binding affinity values calculated suggested that Lp-PLA2 showed broad substrate specificity (Supplementary Fig. 12).

Subsequently, we directly tested the involvement of Lp-PLA2 in the cleavage of PE-18:0_20:4-d11 using lysates from WT and *PLA2G7* KO cells by tracing AA-d11, a product of Lp-PLA2. Although WT cell lysates

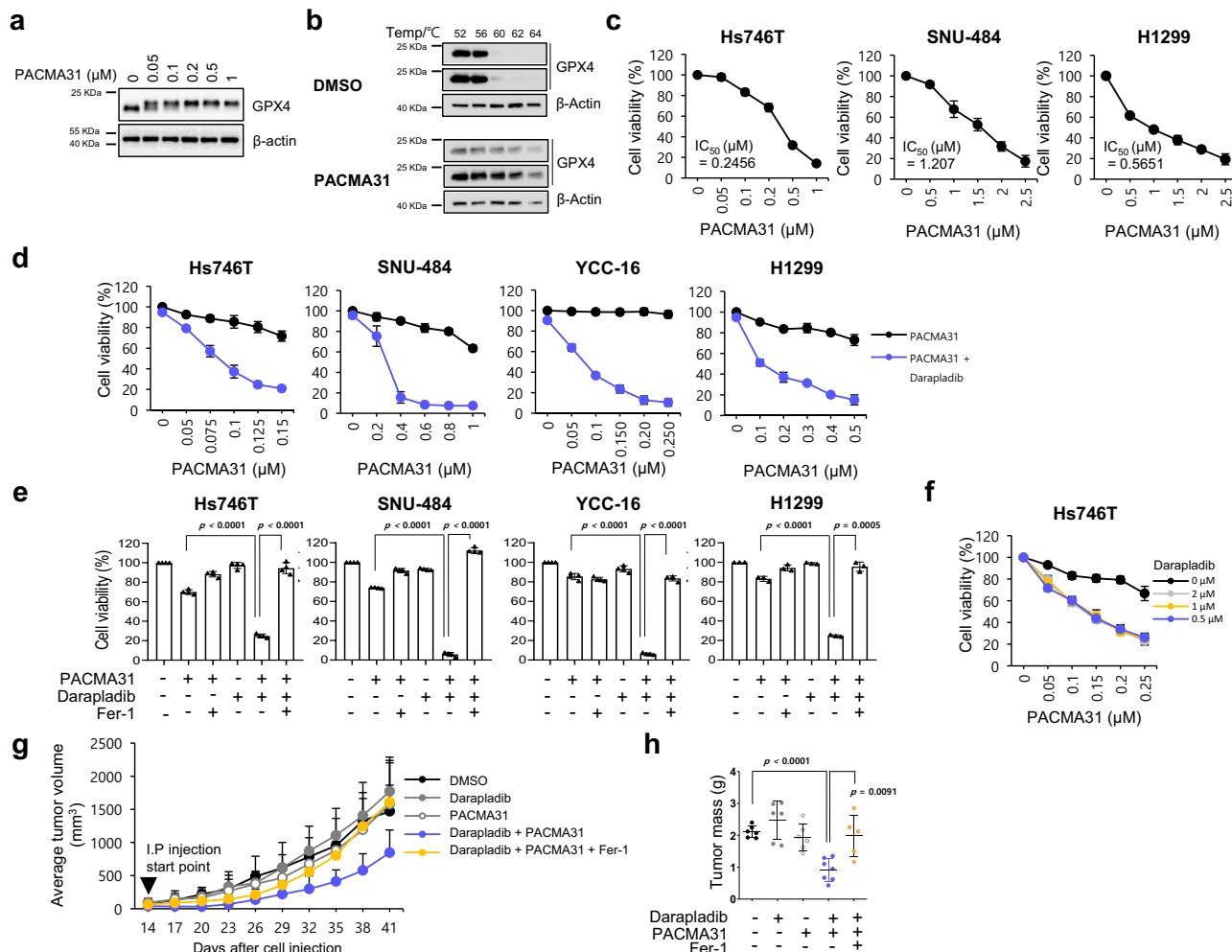

**Fig. 7 | Darapladib enhances the antitumour activity of PACMA31 by accelerating ferroptosis. a** Western blot showing the band shift of GPX4 after PACMA31 treatment. Experiments repeated three times. **b** Cellular thermal shift assay showing the thermal stabilisation of GPX4 in cells treated with PACMA31. Experiments were repeated three times. **c** Relative viability and IC₅₀ value of Hs746T, SNU-484, and H1299 cells treated with PACMA31. The data are presented as the means ± SDs (Hs746T: $n = 6$, SNU-484: $n = 4$, H1299: $n = 3$ independent experiments). **d, e** Relative viability of Hs746T, SNU-484, YCC-16 and H1299 cells treated with PACMA31 and/or 2 μM darapladib in the presence or absence of Fer-1 for 20 h. The data are presented as the means ± SDs ($n = 3$–6 independent experiments, the

significance of the results was assessed using a two-tailed Student's $t$ test). **f** Relative viability of Hs746T cells treated with PACMA31 and/or various concentrations of darapladib. The data are presented as the means ± SDs ($n = 3$ independent experiments). **g, h** SNU-484 cells were injected subcutaneously into nude mice; 14 days after inoculation, the mice were treated with 10 mg/kg PACMA31, 10 mg/kg darapladib, and 10 mg/kg fer-1 via intraperitoneal injections every 3 days as indicated. Tumour growth is shown in (**g**), and tumours are shown in (**h**). Each treatment group consisted of 5–7 mice. The data are presented as the means ± SDs, the significance of the results was assessed using a two-tailed Student's $t$ test). Exact $p$ values provided as source data. Source data are provided as a source data file.

efficiently produced AA-d11 from PE-18:0_20:4-d11, lysates from *PLA2G7* KO cells exhibited a reduced ability to cleave PE-18:0_20:4-d11, indicating that Lp-PLA2 contributes to the PE deacylation cycle (Supplementary Data 8, Supplementary Fig. 11c). Furthermore, darapladib strongly abolished the production of AA-d11, whereas (S)-BEL, an inhibitor of iPLA2, showed a limited ability, suggesting that Lp-PLA2 activity is needed for PE cleavage (Supplementary Data 4, Fig. 6c). These data suggest that Lp-PLA2 inhibition renders cells sensitive to ferroptosis by increasing the levels of PE-18:0_20:4 and protecting oxPE-18:0_20:4 from cleavage.

We have previously shown that supplementation with AA ultimately increases the PE-18:0_20:4 levels and thereby sensitises cells to ferroptosis[8]. Because our results suggest that darapladib induces ferroptosis by ultimately increasing the levels of PE-18:0_20:4, we wondered whether darapladib could no longer promote ferroptosis in the presence of a high concentration of AA. As expected, AA increased the susceptibility of cells to ferroptosis, but darapladib only slightly exacerbated ferroptosis in the presence of AA (Fig. 6d). In contrast,

supplementation with OA, the levels of which were decreased upon darapladib treatment, suppressed ferroptosis in the presence of RSL3 and darapladib (Fig. 6e). These data support the notion that both the increase in PE-18:0_20:4 and the decrease in OA are responsible for ferroptosis sensitisation by darapladib.

## PLA2G7 is upregulated in various types of tumours, and its inhibition with PACMA31 effectively suppresses gastric tumour growth

Although Lp-PLA2 encoded by PLA2G7 genes is known to be mainly expressed in monocytes and macrophages[57], its overexpression in tumours, including prostate, colon, and breast cancer, has been observed in a number of studies[49,58–62]. We therefore investigated the expression of Lp-PLA2 in tumours compared to normal tissues in The Cancer Genome Atlas (TCGA) and found that Lp-PLA2 is upregulated in several types of cancer, including gastric cancer, oesophageal cancer, lung adenocarcinoma, and breast cancer (Supplementary Fig. 13a). Since we focused on the function of Lp-PLA2 in gastric cancer,

we further analysed the expression of Lp-PLA2 in several independent gastric cancer cohorts and observed the general overexpression of Lp-PLA2 in tumours compared to normal tissues (Supplementary Fig. 13b).

We next wondered whether darapladib also contributes to the antitumour effect of GPX4 inhibitors. Because most GPX4 inhibitors, including RSL3 and ML210, are not suitable for in vivo use[10], we employed PACMA31, a previously known PDI inhibitor that was recently identified as a new GPX4 inhibitor and is available for in vivo study[48,63]. We first confirmed that PACMA31 induced a band shift of GPX4 in a dose-dependent manner similar to other GPX4 inhibitors and increased the thermal stability of GPX4 in cell lysates, suggesting that PACMA31 directly binds to GPX4 (Fig. 7a, b). In addition, PACMA31 exhibited IC50 values comparable to those of other GPX4 inhibitors in cells, and the effects were completely inhibited by ferrostatin-1, supporting the notion that PACMA31 is a reliable ferroptosis inducer (Fig. 7c and Supplementary Fig. 14a). Strikingly, PACMA31 killed cells synergistically with darapladib (Fig. 7d–f).

We then tested the combination treatment of PACMA31 and darapladib in a mouse xenograft model established using SNU-484 cells. SNU-484 cells were injected subcutaneously into nude mice, after which the mice were treated with PACMA31 and/or darapladib. Treatment of mice with PACMA31 or darapladib alone showed no statistically significant difference compared to treatment with the vehicle control (Fig. 7g). However, tumour growth in mice treated with both PACMA31 and darapladib was drastically reduced compared to that in the other groups. Eventually, the size and weight of the tumours were significantly reduced in the combination treatment group compared to the other groups (Fig. 7h). Furthermore, ferrostatin-1 rescued the reduced tumour growth and weight after darapladib and PACMA31 treatment, suggesting that ferroptosis contributes to tumour suppression (Fig. 7g, h). Our investigation of the lipid profiles in xenografted tumours treated with PACMA31 and/or darapladib revealed a general increase in PE species in darapladib-treated xenograft tumour tissues, but this increase was not as pronounced as in the cells treated with darapladib (Supplementary Fig. 14b). Taken together, these findings suggest that inhibition of both Lp-PLA2 and GPX4 synergistically inhibits tumour growth in vivo.

## Discussion

The critical roles of ferroptosis in various human diseases, including neurological disorders, ischaemia–reperfusion injury, kidney damage, and blood disorders, have become widely recognised through extensive in vitro and in vivo studies[12,50,64–67]. In addition, cancer cells, especially chemoresistant or persister cancer cells, are highly vulnerable to ferroptosis inducers, which suggests a strategy for anticancer therapy using ferroptosis[8–11]. Most cells implement several protection mechanisms against ferroptosis; for example, the GPX4/GSH system primarily removes lipid peroxides, acting as a first line of defence. Therefore, inhibition of the GPX4/GSH system is necessary to induce ferroptosis in most contexts. NRF2 pathways generally suppress ferroptosis in various contexts[33,34]. In addition, FSP1/AIFM2 and DHODH regenerate CoQ10, allowing CoQ10 to directly neutralise lipid peroxyl radicals[68–70]. Furthermore, iPLA2β can directly cleave oxidised SAPE (HpETE-PE)[24–26]. Finally, the endosomal sorting complexes required for transport (ESCRT) machinery ultimately protects against cell membrane destruction upon ferroptosis[71,72].

While AA and AdA are the most susceptible PUFAs to peroxidation, how these lipids are controlled in cells is largely unknown[3]. The most abundant PUFA in serum and plasma is LA (C18:2), and a comparable amount of AA is also found in serum[73]. However, very long-chain PUFAs (VLC-PUFAs) with more than 22 carbons, such as AdA (C22:4), are rarely present in serum and plasma. In this regard, we recently reported that cell-autonomous synthesis of AA and AdA by fatty acid desaturases (FADSs) and elongases (ELOVLs) is critical for ferroptosis, although AA can be imported when PUFA synthesis is inhibited[8]. Nevertheless, a number of studies have suggested that AA import and AA esterification by ACSL4 play a pivotal role in ferroptosis[17,74,75]. Therefore, precise studies are needed to determine how much each pathway contributes to the intracellular levels of AA under various conditions. The results of this study suggest that the cell-autonomous recycling of AA by Lp-PLA2 is a crucial pathway that determines the amount of AA-containing PE and PE-p and thereby contributes to ferroptosis resistance. In addition, this pathway might be favourable for a rapid response to ferroptotic stimuli because PUFA synthesis or uptake seems to be a relatively slow process compared to PUFA recycling by Lp-PLA2.

While PLA2 superfamily members have the same fundamental function, their functions may differ in different contexts, such as in different subcellular locations and tissues. Lp-PLA2 is unique in that it mainly binds to lipoproteins outside cells and acts on phospholipids in lipoproteins. The most well-defined role of Lp-PLA2 is its contribution to atherosclerosis. Lp-PLA2 is strongly expressed in the necrotic core and cleaves oxLDLs, mostly oxPC, into lysoPCs and NEFAs, leading to macrophage death and thereby contributing to atherosclerosis[76]. However, apart from its classical role, Lp-PLA2 seems to suppress ferroptosis independently of lipoprotein because *PLA2G7* KO cells or cells treated with the Lp-PLA2 inhibitor darapladib were more sensitive to ferroptosis under lipoprotein-deficient or serum-deficient conditions (Figs. 3 and 5). In addition, the Lp-PLA2 inhibitor did not reduce the extracellular levels of lysoPC (Supplementary Fig. 6). Although we could not completely exclude the possible off-target effect of darapladib on the promotion of ferroptosis, the observation that Lp-PLA2 depletion or deletion also augments ferroptosis supports the idea that Lp-PLA2 is an important regulator of ferroptosis.

Lp-PLA2 cleaves not only oxidised LDL but also oxidised PS, an eat-me signal of apoptotic cells, and thereby restricts phagocytosis by macrophages[54]. A recent study suggested that Lp-PLA2 also recognises and cleaves oxPE-(C18:0_C20:4) upon ferroptosis stimuli, implying that oxPE-(C18:0_C20:4) acts as an "eat-me" signal of ferroptotic cells that is recognised by macrophages[56]. However, whether these activities are related to ferroptotic cell death has not been studied. We also observed that the ratio between oxPE-(C18:0_C20:4) and PE-(C18:0_C20:4) was slightly increased by the inhibition of Lp-PLA2 (Fig. 6 and Supplementary Fig. 11). However, we also observed global rearrangement in the lipid profile upon the inhibition of Lp-PLA2. In particular, Lp-PLA2 seemed to exert PLA2 activity, as several phospholipids, such as PE, PI, and PS, accumulated while their corresponding lysophospholipids were downregulated upon Lp-PLA2 inhibitor treatment. Because the binding affinity suggested by our molecular docking simulation was relatively low, there may be additional mechanism that determine the specificity of Lp-PLA2 in cells, and thus the direct determination of the tertiary structure of Lp-PLA2 and phospholipids in condition similar to the cell membrane may be required.

Interestingly, PC, the most abundant phospholipid, was not significantly affected by the Lp-PLA2 inhibitor. These observations raise the possibility that Lp-PLA2 may also affect the Kennedy pathway, which is involved in the conversion of PC from PE. However, we observed these lipidomic changes within 1 h, which seemed to be insufficient time for the lipidome to be changed through the Kennedy pathway. Rather, lysoPCs abundant in the medium were rapidly scavenged by the cells and used for PC synthesis, thereby contributing to maintaining intracellular PC levels under normal conditions. Although Lp-PLA2 inhibition may protect against PC cleavage in cells, it may also delay PC synthesis because the reduction in lysoPCs in the medium was slightly inhibited by darapladib (Supplementary Fig. 5). In addition to PE species, OA, the most abundant MUFA in serum that prevents ferroptosis, was downregulated by darapladib treatment, possibly contributing to the increased susceptibility to ferroptosis. Interestingly, while we did not observe drastic changes in lysoPC in the medium, supplementation with lysoPCs, such as lysoPC16:0 and lysoPC18:1,

abrogated RSL3-induced ferroptosis, although the underlying mechanism needs further investigation (Supplementary Fig. 15).

Because PLA2 is known to specifically cleave phospholipids at the sn-2 position into FFAs and lysophospholipids (lysoPLs)[77], other PLA2 isoforms may control ferroptosis by regulating PUFA-containing PL abundance. Although the depletion of several PLA2 isoforms resulted in context-dependent promotion of ferroptosis, the inhibition of each PLA2 isoform with their inhibitors had no obvious effect on ferroptosis in cancer cells (Supplementary Fig. 16a, b). Interestingly, (S)-BEL, an inhibitor of iPLA2, is itself toxic to H1299 cells at high concentrations but slightly enhances ferroptosis, possibly due to its ability to inhibit iPLA2 activities against oxidised and non-oxidised SAPE (Supplementary Fig. 16c)[24,25]. Notably, darapladib sensitised various cell lines, including H9c2 cells, to ferroptosis, suggesting that Lp-PLA2 may be a common regulator of ferroptosis that conserves PEs containing PUFAs.

The induction of ferroptosis is an attractive strategy for cancer therapy because a single inhibitor of GPX4 or SLC7A11 can markedly induce ferroptosis in cells. However, with its poor pharmacokinetic profile, the use of RSL3 in vivo is very limited, and only the antitumour activity of RSL3 in vivo has been tested by intratumoural injection as a proof-of-concept[78]. In this study, we employed PACMA31, a recently identified GPX4 inhibitor that was previously known as a PDI inhibitor[48,63]. We also confirmed that PACMA31 has high in vitro potency to kill cancer cells and directly binds to GPX4, as evidenced by a band shift. In combination treatment, PACMA31 and darapladib exhibited strong antitumour activity in a mouse xenograft model. However, due to the low in vivo potency and the lack of oxidised lipid analysis, whether PACMA31 is a reliable ferroptosis inducer in mice requires further investigation. Since GPX4 inhibition may be toxic to various normal tissues, including the heart, kidneys, and liver, how to deliver drugs specifically to cancer tissues remains a challenge. Because no toxicity issues were observed in phase III clinical trials, darapladib may be used as a repositioning drug for cancer therapy. In summary, Lp-PLA2 reduces the levels of PUFA-containing PEs and cleaves oxidised PEs, thereby suppressing ferroptosis. Therefore, inhibition of Lp-PLA2 using darapladib may be beneficial for ferroptosis-inducing strategies for cancer therapy. Our findings provide new insights into possible combination strategies targeting both Lp-PLA2 and ferroptosis in cancer.

## Methods

### Cell culture
Hs746T (30135), SNU-484 (00484), and YCC-16 cells were purchased from the Korean Cell Line Bank (KCLB) at Seoul National University, and H1299 (CRL-5803), A549 (CCL-185), HepG2 (HB-8065), HEK293T (CRL-3216) and H9c2 (CRL-1446) cells were purchased from the American Type Culture Collection (USA). Hs746T, H1299, A549, HepG2, HEK293T and H9c2 cells were cultured in Dulbecco's modified Eagle's medium (DMEM) (HyClone). SNU-484 and H1299 cells were cultured in RPMI medium (Corning). YCC-16 cells were cultured in minimal essential medium (MEM) (HyClone). Lipoprotein-deficient serum from human plasma (S5519) was purchased from Sigma-Aldrich (USA). All media were supplemented with 10% fetal bovine serum (Gibco) and 100X antibiotics (Gibco). All of the cells were incubated at 37 °C in a humidified atmosphere with 5% $CO_2$. None of the cell lines used fall under the category of commonly misidentified cell lines. Detailed information on materials and cells used in our study is presented in Supplementary Table 4.

### Western blotting and antibodies
Cells were lysed using Nonidet-P40 lysis buffer (1% NP-40, 150 mM NaCl, 10% glycerol, 1 mM EDTA, and 20 mM Tris-HCl [pH 7.4]). Then, the membranes were blocked with 5% skim milk in TBS-Tween 20 (TBST), incubated with primary antibodies overnight at 4 °C and then incubated with secondary antibodies for 1 h at room temperature.

For the detection of ELOVL5 protein, the samples were not boiled as previously described[8]. Primary antibodies against ACC (3662, 1:1000), ACC-p (3661,1:1000), FASN (3180,1:1000), FTH1 (3998,1:1000), S6K (9202,1:1000) and S6K-p (9205,1:1000) were purchased from Cell Signaling Technology (USA). SREBP1 (557036,1:1000) was purchased from BD Biosciences (USA). ELOVL5 (sc-398653,1:2000), ACSL4 (sc-365230,1:1000), FSP1 (sc-377120,1:1000), PEBP1 (sc-376925,1:2000) and HSP90 (sc-13119,1:3000) were procured from Santa Cruz (USA). GPX4 (ab125066,1:2000) and Na/K ATPase (ab76020,1:1000) were purchased from Abcam (UK). NRF2 (16396-1-AP,1:1000) and FADS1 (10627-1-AP,1:1000) were obtained from Proteintech (USA). LPCAT3 (16-999,1:1000) was purchased from ProSci (USA). Flag (F3165,1:3000), β-actin (A5316,1:10,000) and α-tubulin (T9026,1:10,000) were purchased from Sigma-Aldrich (USA). Detailed information on materials and cells used in our study is presented in Supplementary Table 4.

### Reagents
Darapladib (S7520), Fer-1 (S7243), RSL-3 (S8155), zVAD-fmk (zVAD, S7023) and erastin (S7242) were purchased from Selleck Chemicals (USA). PACMA31 (5116) was purchased from Tocris (UK). Liproxstatin-1 (SML1414), LDL (L7914), HDL (L8039), VLDL (437647),1-stearoyl-2-arachidonoyl-sn-glycero-3-PE (PE 18:0_20:4) (850804C) and ML210 (SML0521) were procured from Sigma-Aldrich (USA). Nec-1 (BML-AP309) was purchased from Enzo Life Sciences. MAFP (70660), S-BEL (70700), MJ33 (90001844), JKE1674 (30784), AA (90010), OA (90260) and 1-stearoyl-2-arachidonyl-d11-sn-glycero-3-PE (SAPE-d11) (27929) were obtained from Cayman Chemical (USA). Detailed information on materials and cells used in our study is presented in Supplementary Table 4.

pCMV3-SP-N-FLAG-Lp-PLA2 (HG10848-NF) was purchased from Sinobiological (China) and subcloned into the pCMV2-FLAG vector.

### Cell viability and LDH release assays
Cell viability was determined by measuring the cellular ATP levels using CellTiter-Glo reagent (CellTiter-Glo® 2.0 Assay, Promega) according to the manufacturer's protocol. Cells were seeded at a density of $3 \times 10^4$ cells/well in 200 μl of culture medium in 48-well plates. For low density, $2 \times 10^4$ cells/well were seeded in 48-well plates. The day after seeding, chemicals were administered at the concentrations and times indicated in the text. A volume of solution equal to that of the culture medium was added, and the cells were incubated at room temperature for 30 min. Then, 150 μl of the supernatant was transferred to a 96-well plate, and the absorbance was measured at 450 nm using a microplate reader. Cell-free supernatants were collected and assessed via LDH release assay with a Cytotoxicity Detection Kit (Roche). The percent LDH release was calculated according to the manufacturer's instructions.

### Crystal violet staining
A total of $1 \times 10^5$ cells were cultured in 24-well plates and exposed to drugs for 20 h. After the medium was removed, the cells were washed in cold phosphate-buffered saline (PBS), fixed with 100% methanol, and stained with 0.25% crystal violet solution.

### PI staining
The cells were seeded into 12-well plates at a density of $1 \times 10^5$ cells/well and treated with the indicated concentrations of drugs for 20 h. Propidium iodide (PI, 10 μg/ml) was then added to each well, and the plates were incubated at 37 °C for 15 min in the dark. The cells were harvested by two washes with PBS, transferred to a 1.5 ml microfuge tube (800 × g, 3 min), and then resuspended in 0.5 ml of PBS for flow cytometry. The cells were analysed using the 595nm laser of a flow cytometer (BD FACS Calibur) for excitation, and data were collected from the FL2 detector. A minimum of 10,000 cells were analysed per condition. BD Cellquest Pro (BD bioscience, USA) software was used for data acquisition and analysis.

## Cysteine deprivation

For cysteine deprivation, cysteine-free DMEM (lacking glutamine, methionine, and cysteine; Gibco) supplemented with methionine, 10% dialysed FBS (26400044; Gibco) and 2 mM L-glutamine (Gibco) was used. The cells were washed with PBS and cultured with fresh DMEM or cysteine-free DMEM for 18 h in the presence or absence of Fer-1.

## Lipid peroxidation assay

Lipid peroxidation was measured with C11-BODIPY$^{581/591}$ (BODIPY™ 581/591 C11, Invitrogen). When lipid peroxidation increases, the fluorescence shifts from red to green. Cells were treated with RSL3 and/or darapladib and incubated for the indicated times. Next, 2.5 μM C11-BODIPY$^{581/591}$ was added, and the cells were incubated at 37 °C for 15 min. The cells were harvested by washing twice with PBS, transferred to a 1.5 ml microfuge tube (800 × $g$, 3 min), and then resuspended in 0.5 ml of PBS for flow cytometry. The cells were analysed using the 480nm laser of a flow cytometer (BD FACS Calibur) for excitation, and data were collected from the FL1 detector. A minimum of 10,000 cells were analysed per condition. BD Cellquest Pro (BD bioscience, USA) software was used for data acquisition and analysis.

## Iron assay

The iron concentration was determined using an iron colorimetric assay kit (Sigma). According to the manufacturer's instructions, cells were harvested, washed with ice-cold PBS, homogenised in 5X volumes of Iron Assay Buffer and centrifuged at 16,000 × $g$ for 10 min at 4 °C to obtain the supernatant. Five microlitres of iron reducer was added to each of the sample (50 μl/well) wells to reduce $Fe^{3+}$ to $Fe^{2+}$ and incubated for 30 min at 25 °C. Subsequently, 100 μl of Iron Probe was added to each well, and the reaction was incubated in the dark for 60 min at 25 °C. The absorbance at 593 nm was detected using a microplate reader. The absorbance values were calibrated to a standard concentration curve to calculate the concentration of iron.

## GPX4 knockdown cell lines

Lentiviral vectors including GPX4 shRNA (TRCN number_46251, 46249, 46252) and negative control were purchased from Sigma-Aldrich. For lentivirus production, HEK293T packaging cells were transfected with 12 μg of lentiviral vectors, including GPX4 shRNA, 6 μg psPAX2 and 3 μg pMD2.G using Lipofectamine 3000 transfection reagent. Forty-eight hours after 48, the viral supernatant was collected and filtered. H1299 cells were incubated for 5 h with the viral supernatant and supplemented with 10 μg/ml polybrene. Puromycin (5 μg/ml) and Fer-1 (2 μM) were used to select the stable cell line. The GPX4-shRNA interference effect was examined by Western blotting.

## RNAi-mediated gene knockdown

An ON-TARGET plus SMARTpool for human Lp-PLA2 siRNA (L-004903-00-0020), PA2G4A siRNA (L-0009886-00-0005), PLA2G2A siRNA (L-009901-00-0005), PLA2G6 siRNA (L-009085-00-0005) and a nontargeting pool (siNT, D-001810-10) were purchased from Dharmacon (USA). Cells were transfected with 20 nM siRNA using Lipofectamine RNAiMAX (Invitrogen) according to the manufacturer's protocol.

## PLA2G7 knockout cell lines

PLA2G7 KO cell lines were produced by homologous recombination with the CRISPR−Cas9 system. The CRISPR−Cas9 constructs of PLA2G7 (LP-PLA2) were obtained by ligation of double-stranded antisense/sense oligos to the short guide RNAs (sgRNAs) of interest in pSpCas9(BB)−2A-GFP (PX458; Addgene #48138) and pSpCas9(BB)−2A-RFP plasmids. The CRISPR vector was created based on the manufacturer's protocol. The CRISPR plasmid was cotransfected into cells using Lipofectamine 3000 transfection reagent (L3000015, Invitrogen). Forty-eight hours following transfection, green and red fluorescent protein-positive cells were selected by FACS and seeded into 96-well plates, and single clones were expanded. The insert sequences of the constructs are as follows:

 sgRNA1-PLA2G7-S: CACCGGACCAATCTGCTGCAGAAAT,
 sgRNA1-PLA2G7-AS: AAACATTTCTGCAGCAGATTGGTCC,
 sgRNA2-PLA2G7-S: CACCGGCAGTAATTGGACATTCTTT, and
 sgRNA2-PLA2G7-AS: AAACAAAGAATGTCCAATTACTGCC.

## Total RNA extraction and RT−qPCR

Total RNA was extracted using TRIzol (Gibco) according to the manufacturer's protocol. First-strand complementary DNA (cDNA) synthesis was performed using 2 μg of total RNA with M-MLV reverse transcriptase (Promega) according to the manufacturer's protocol. All RT−qPCR was performed with SYBR Premix Ex Taq™ (Takara Bio Inc., Japan) using a CFX96™ Real-Time System (Bio-Rad Laboratories, USA).

The primer sequences were as follows: Lp-PLA2 (forward), 5′-TGG CTCTACCTTAGAACCCTGA-3′; Lp-PLA2 (reverse), 5′-TTTTGCTCTTTG CCGTACCT-3′; FSP1 (forward), 5′-TCCGTCCGGCAGGAAGTGAA-3′; FSP1 (reverse), 5′-CATTGAGAGGCAGCTCCTCC-3′; GSTA (forward), 5′-AGCCCAAGCTCCACTACTTC-3′; GSTA (reverse), 5′- CTTCAAACTCTAC TCCAGCTGCAG-3′; HO-1 (forward), 5′-GCCAGGTGCTCAAAAAGATT-3′; HO-1 (reverse), 5′- CCTGCAACTCCTCAAAAGAGC-3′; Ferritin heavy chain (FTH)−1 (forward), 5′-GTTGTACCAAAACATCCACTTAAG-3′; FTH-1 (reverse), 5′-CCTCAAAGACAACACCTGGG-3′; NQO1 (forward), 5′- CGTTTCTTCCATCCTTCCAG-3′; NQO1 (reverse), 5′-CGCAGACCTT GTGATATTCC-3′; β-Actin (forward), 5′-CTGGCACCCAGCACAATG-3′; and β-Actin (reverse), 5′-GCCGATCCACACGGAGTACT-3′.

## Cellular thermal shift assay (CETSA)

A cellular thermal shift assay was conducted as previously described[79]. Briefly, Hs746T cells were treated with 1 μM PACMA31 for 1 h and incubated at increasing temperatures for 3 min. The cells were then lysed using the freezing and thawing method, and the soluble fraction obtained after centrifugation was subjected to Western blot analysis to confirm the thermal stability of GPX4.

## Lipidomic profiling by UPLC/QTOF MS

For global lipid profiling, the lipids in cells, culture media and tumour tissues were analysed by ultra-performance liquid chromatography (UPLC) coupled with quadrupole time-of-flight mass spectrometry (QTOF-MS). For intracellular lipid analysis, 3 to $5 \times 10^5$ cells were washed twice with PBS and extracted by scraping into 300 μl of methanol:acetonitrile:water 40:40:20 (v/v) containing 0.5% formic acid. The lipid extracts were cleared by centrifugation at 18,500 × $g$ for 15 min at 4 °C, and the supernatants were transferred to vials for lipidomic analysis. Extracellular lipids were extracted by adding 800 μl of methanol:acetonitrile 50:50 (v/v) containing 0.625% formic acid to 400 μl of culture medium and centrifuging the mixture. Eight hundred microlitres of supernatant was transferred and dried under nitrogen gas. The medium extract was reconstituted in 80 μl of methanol:acetonitrile:water (40:40:20, v/v/v). For lipid extraction from tumours, 25 to 220 mg of tumour was mixed with cold chloroform:methanol (1.75:1, v/v) solution at a ratio of 1:4.7 (ratio of tumour weight). The mixture was then homogenised twice with 2.8-mm zirconium oxide beads at 2800 × $g$ for 25 s using a Precellys 24 tissue grinder (Bertin Technologies, France) and incubated at −20 °C for 50 min. Water was added at a ratio of 1:4.7 (ratio of tumour weight), and the resulting mixture was vortexed and stored at 4 °C for 15 min. After centrifugation at 34,300 × $g$ and 4 °C for 20 min, the organic phase was transferred to a new EP tube and dried under nitrogen gas. Lipid extracts were reconstituted with isopropanol:acetonitrile:water (50:25:25, v/v/v), and the solvent volume (μl) was 5-fold higher than the tumour weight (mg). Tridecanoic acid (0.5 μM, FFA 13:0, Sigma-Aldrich, USA) and a 50-fold dilution of SPLASH Lipidomix (Avanti Polar Lipids, USA) were mixed with the extraction solution as internal standards. Intracellular

and extracellular lipid analysis was conducted using an ACQUITY UPLC system (Waters, Milford, MA, USA) coupled with a triple TOF1™ 5600 mass spectrometer equipped with an electrospray ionisation (ESI) source (AB Sciex, Concord, ON, Canada). Chromatographic separation of lipids in the cells and media was performed using an ACQUITY UPLC CSH C18 column (2.1 mm × 100 mm, 1.7 μm; Waters) at 55 °C and a flow rate of 0.4 ml/min. The binary mobile phases comprised 10 mM ammonium acetate in water:acetonitrile (60:40 v/v, solvent A) and isopropanol:acetonitrile (90:10 v/v, solvent B). The gradient was as follows: 40–43% B from 0–2 min, 43–50% B from 2–2.1 min, 50–54% B from 2.1–12 min, 54–70% B from 12–12.1 min, 70–99% B from 12.1–18 min, 99–40% B from 18–18.1 min, and 40% B from 18.1–20 min to equilibrate. The mass spectrometer was operated in negative ion mode, and the mass range was set at 80–1500 $m/z$ for analysis of the lipid extracts. Total ion chromatograms were acquired using the following operating parameters: capillary voltage of −4500 V, nebuliser pressure of 50 psi, drying gas pressure of 60 psi, curtain gas pressure of 30 psi, source temperature of 500 °C, declustering potential of −90 eV, collision energy of −10 eV for single MS, and collision energy of −45 eV for MS/MS. Data from MS/MS analyses were acquired using automatic fragmentation, in which the five most intense mass peaks were fragmented. Mass accuracy was maintained with an automated calibrant delivery system interfaced to the second inlet of the DuoSpray ion source. All the samples were pooled in equal volumes to generate quality control (QC) samples, which were analysed prior to sample acquisition and after every 7 samples. Spectral data were preprocessed using MarkerView™ software (version 1.3.1, AB Sciex) and normalised to the total spectral area. Lipidomic analysis of tumours was performed using a Xevo G2-XS Q/TOF mass spectrometry (MS) system coupled with an ultra-performance liquid chromatography (UPLC) system (Waters, Milford, MA, USA). The eluate was analysed by electrospray ionisation (ESI) in the negative mode. The mass range was 50–1200 $m/z$. Leucine-enkephalin ([M-H]⁻: $m/z$ 554.2615) was used as a reference in the lock-spray and introduced by a lock-spray at 5 μl/min for accurate mass acquisition. Raw UPLC/QTOF MS spectral data were preprocessed using Progenesis QI software (version 2.0, Waters, Milford, MA, USA). All data were normalised to the detected total lipids. Lipid metabolites were identified by comparing experimental data with online databases (DBs; HMDB, METLIN, and LIPID MAPS) and our in-house library. Identification was confirmed using the MS/MS patterns and retention times of lipid standard compounds. Representative MS/MS spectra of various lipid classes are presented in Supplementary Fig. 17, see also Supplementary Data 1, 2, 9.

## Analysis of AA and lysoPCs in medium

Extracellular lipid extracts were analysed by UPLC/triple-quadrupole mass spectrometry (UPLC/TQ MS) in multiple reaction monitoring (MRM) mode using mobile phases and a column in the same manner as that for lipidomic profiling, but the column temperature was maintained at 30 °C. An Agilent 1290 Infinity II LC and Agilent 6495 Triple Quadrupole MS system equipped with an Agilent Jet Stream ESI source (Agilent Technologies) was used for the analysis of AA and lysoPCs. Mass Hunter Workstation (ver. B.06.00, Agilent Technologies) software was used for data acquisition and analysis. Samples were eluted at 0.2 ml/min for 22 min. Gradient elution was conducted as follows: 40–55% B from 0–6 min, 55–60% B from 6–11 min, 60–99% B from 11–14 min, 99% B from 14–18 min, 99–40% B from 18–18.1 min, and 40% B from 18.1–22 min. MS/MS experiments were conducted in negative ion mode. The MS system was operated using the following parameter settings: gas temperature of 220 °C, nebuliser gas of nitrogen at 30 psi, sheath gas temperature of 300 °C, and sheath gas flow rate of 11 l/min. Individual standards were used to optimise the conditions of each lipid, and the transitions and retention times are summarised in Supplementary Table 1. Subsequently, 0.5 μM FFA 13:0 and 500 ppb lysoPC 18:1-d₇ (Avanti Polar Lipids, USA) were mixed with

methanol:acetonitrile (50:50, v/v) solution, and the extraction process was the same as that used for lipidomic analysis. The intensities were normalised to the cell numbers.

## Analysis of oxidised PE and PE in cells

For the analysis of oxidised PE, PE-(18:0_20:4), and PE-(18:0_22:4) in cells, the lipids were extracted in 500 μl of methanol:water 80:20 (v/v) and 800 μl of chloroform. The lipid extracts were dried under nitrogen gas and reconstituted in isopropanol/acetonitrile/water 50:25:25 (v/v/v). A total of 100 ppb PE-15:0/18:1-d₇ (Sigma-Aldrich, USA) was mixed with methanol:water (80:20, v/v) or redissolution solvent as an internal standard. The individual concentrations, including those of PE-18:0_20:4 + 2O, PE-18:0_20:4 and PE-18:0_22:4, were normalised to the protein levels. Targeted analysis of oxidised PE and PE was performed with an ACQUITY UPLC I-Class PLUS (Waters, Milford, MA, USA) and a Xevo TQ-XS Triple Quadrupole MS system (Waters, Milford, MA, USA) in the MRM mode. MassLynx (Waters, Milford, MA, USA) software was used for data acquisition and analysis. The mobile phases and chromatographic separation were the same as those used for AA and lysoPC analyses. The MS settings were as follows in the negative mode: gas temperature of 600 °C, nebuliser gas flow at 7 bar, sheath gas temperature of 150 °C, and sheath gas flow rate of 150 l/h. Oxidised PE and PE-18:0/20:4 standards were used for optimization of the conditions for each metabolite and for preparation of a series of calibration solutions to generate calibration curves. Details of the measurement are included in Supplementary Table 2.

## LC−MS-based substrate assay

To perform the lipid substrate assay, H1299 cells were lysed in assay buffer A (30 mM Tris-HCl (pH 7.4), 5 mM EDTA and 120 mM NaCl) and complete protease inhibitor (EDTA-free, Roche) using a probe sonicator (Branson Sonifier model 250) with eight pulses (30% duty cycle, output setting = 4), and clear cell lysate was obtained by centrifugation at 18,500 × $g$ for 20 min at 4 °C. Total protein concentrations were determined using the Bio-Rad Protein Assay Kit. The cell lysates were diluted to 1 mg/ml using assay buffer A. A 100 μg/ml stock solution of lipids (PE(18:0/20:4-d₁₁); 128.35 μM) was prepared by drying the stocks under a nitrogen stream and then resuspending them in assay buffer A. The lipid substrate stock was diluted to 14 μM using assay buffer A. Then, 60 μl of cell lysates, darapladib or (S)-BEL and 190 μl of lipid substrate were combined into a final volume of 250 μl per reaction and incubated at 37 °C with shaking at 200 rpm for 180 min. Additionally, an additional reaction was conducted for 60 min to investigate the involvement of Lp-PLA2 in the cleavage of PE using WT and *PLA2G7* KO lysates. The reaction was quenched with 450 μl of cold chloroform/methanol (2:1, v/v) containing 0.1% formic acid and 35 nM FFA 13:0 as an internal standard. The mixture was vortexed and centrifuged at 16,000 × $g$ for 10 min at 4 °C. After centrifugation, the organic phase was transferred to a new EP tube and dried under nitrogen gas. The lipid extract was reconstituted into 85 μl of isopropanol:acetonitrile:water (50:25:25, v/v/v) and analysed using a Xevo TQ-XS Triple Quadrupole mass spectrometry (MS) system coupled with a Waters ACQUITY UPLC I-Class PLUS system (Waters, Milford, MA, USA). The mobile phases and chromatographic separation were the same as those used for AA and lysoPC analysis, and the parameters of the MS/MS experiments were the same as those used for oxidised PE and PE. Detailed information on the measurements is presented in Supplementary Table 3.

## Protein–ligand docking and structural analysis

The initial structural model for docking simulation was prepared based on the crystal structure of human Lp-PLA2 (PDB ID 6M06)[80] using PyMOL software (version 1.8)[81]. Energy minimisation and protein preparation were then performed using Modeller software (version

10.4)[82]. The 2D structures of lipids, namely, PE-18:0_20:4-OOH, PE-18:0_20:4, PS-18:0_20:4, PI-18:0_20:4, PL-18:0_20:4, and PI-18:0_20:4, were obtained from PubChem DB[83], and their 3D structures for docking studies were generated using MarvinView software (version 23.5.0) and Chemaxon (https://www.chemaxon.com). The initial docking simulation was performed using AutoDock Vina (version 1.2.0)[84] with a box size of X = 37.7, Y = 48.4, and Z = 54.9 and centre X = 205.1, Y = 16.4, and Z = 111.44 and calculated by HDOCK[85].The parameters of exhaustiveness, number of modes, and energy range were set to 8, 10, and 3, respectively. The docking scores were recalculated using PRODIGY[86], and the binding affinities were calculated.

### TCGA Pan-Cancer data and microarray data for gastric cancer analysis

Publicly available data from The Cancer Genome Atlas (TCGA) were analysed in the current study. Clinical information and gene expression RNA-seq data, which were batch-effects-normalised mRNA data ($n = 11,060$), from the TCGA Pan-Cancer (PANCAN) dataset were downloaded from the University of California Santa Cruz Xena platform (http://xena.ucsc.edu). The PANCAN dataset included extracted cancer tissue data and normal tissue data for various cancer types, including stomach adenocarcinoma (STAD), oesophageal carcinoma (ESCA), lung adenocarcinoma (LUAD), lung squamous cell carcinoma (LUSC), colon adenocarcinoma (COAD), breast invasive carcinoma (BRCA) and pancreatic adenocarcinoma (PAAD). Microarray datasets for gastric cancer, namely, GSE66229, GSE29272, and GSE27342, were obtained from the Gene Expression Omnibus database (https://www.ncbi.cn) of the National Center for Biotechnology Information. The *PLA2G7* expression data were analysed for the current study, and the $\log_2(\text{norm\_value} + 1)$ values or the $\log_2(\text{norm\_value})$ values for TCGA were used to represent the expression levels of the genes.

### In vivo tumour xenograft studies

SNU-484 cells in the logarithmic growth phase were transplanted into 6-week old female CAnN.Cg-Foxn1 nu/CrlOri mice ($6 \times 10^6$ cells in 200 μl of PBS + Matrigel/mouse). To determine the anticancer effect, the tumour size and weight of the mice were measured every 3 days. The tumour volume was calculated according to the formula tumour volume = longest diameter × shortest diameter × height/2. For intraperitoneal (i.p.) administration, the tumour was allowed to grow for ~2 weeks to reach an average volume of ~20 mm³. The mice were randomly divided into 4 groups: the vehicle control group ($n = 7$), the PACMA31 group ($n = 5$; 10 mg/kg), the darapladib group ($n = 7$; 10 mg/kg), the PACMA31 + darapladib group ($n = 5$) and the PACMA31 + darapladib + Fer-1 group (10 mg/kg) ($n = 5$). The maximal tumour size/burden permitted in the protocol of the institutional animal care and use committees did not exceed 2000 mm³. Mice were housed under 12 light/12 dark cycle, temperatures of $22 \pm 2\,°C$ with $50 \pm 10\%$ humidity. All experimental protocols were approved by the Institutional Animal Care and Use Committee of the Korea Research Institute of Bioscience & Biotechnology (KRIBB, KRIBB-AEC-21014) and were conducted in accordance with the guidelines for the Care and Use of Laboratory Animals published by the US National Institutes of Health.

### Statistics and reproducibility

All experiments were repeated at least three times. The data are presented as the means ± standard deviations (SDs). The statistical significance of the differences among groups were analysed using GraphPad Prism 8 (GraphPad Software) and Excel.

### Reporting summary

Further information on research design is available in the Nature Portfolio Reporting Summary linked to this article.

## Data availability

The mass spectrometry data have been deposited to the ProteomeXchange Consortium via the PRIDE[87] partner repository with the dataset identifier PXD044002. Source data are provided with this paper.

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

## Acknowledgements

This work was supported by a grant from the KRIBB Research Initiative Program and Korea Basic Science Institute (C370000), by the R&D Convergence Program (CAP-21022-000), by grants from the Korea Disease Control and Prevention Agency (2021ER160702), and by National Research Foundation of Korea (NRF) grants funded by the Korean government (MSIT) (Nos. 2017M3A9G5083321 (S.C.L.), 2020R1A2C1006841 (S.C.L.), 2020R1A2C2007835 (G.-S.H.) and 2022R1A2C4002108 (E.-W.L.).

## Author contributions

M.O., S.Y.J., J.-Y.L., S.C.L., B.-S.H., G.-S.H., and E.-W.L. conceived and designed the study and wrote the manuscript; M.O., J.-Y.L., and J.W.K. performed most of the biochemical experiments; S.Y.J., Y.J., Da.K., and G.-S.H. performed the lipidomics experiments; M.O., J.W.K. J.S., and T.-S.H. performed the mouse xenograft experiments; T.-S.H., E.J., and H.Y.S. analysed gene expression from TCGA and the other databases; J.K. and Do.Kim performed the molecular docking analysis; MitoImmune conducted the erastin-induced ferroptosis experiments; and M.W.K., K.-H.S., K.-J.O., W.K.K., K.-H.B., and Y.-M.H. analysed the data and provided useful comments.

## Competing interests

The authors declare no competing interests.

## Additional information

[1]Biodefense Research Center, Korea Research Institute of Bioscience and Biotechnology (KRIBB), Daejeon 34141, Korea. [2]Integrated Metabolomics Research Group, Western Seoul Center, Korea Basic Science Institute, Seoul 03759, Korea. [3]Metabolic Regulation Research Center, Korea Research Institute of Bioscience and Biotechnology (KRIBB), Daejeon 34141, Korea. [4]Department of Functional Genomics, University of Science and Technology (UST), Daejeon 34141, Korea. [5]Therapeutics and Biotechnology Department, Drug Discovery Platform Research Center, Korea Research Institute of Chemical Technology (KRICT), Daejeon 34114, Korea. [6]Graduate School of New Drug Discovery and Development, Chungnam National University, Daejeon 305-764, Korea. [7]Aging Convergence Research Center, Korea Research Institute of Bioscience and Biotechnology (KRIBB), Daejeon 34141, Korea. [8]Biotherapeutics Translational Research Center, Korea Research Institute of Bioscience and Biotechnology (KRIBB), Daejeon 34141, Korea. [9]MediBio-Informatics Research Center, Novomics Co., Ltd., Seoul, Korea. [10]YUHS-KRIBB Medical Convergence Research Institute, Seoul 03722, Korea. [11]Department of Radiology, College of Medicine, Yonsei University, Seoul 03722, Korea. [12]Department of Life Science, Ewha Womans University, Seoul 03760, Korea. [13]MitoImmune Therapeutics Inc., Seoul 06123, Korea. [14]Department of Cell Biology, Daegu Catholic University School of Medicine, Daegu 42472, Korea. [15]College of Pharmacy, Chung-Ang University, Seoul 06974, Korea. [16]School of Pharmacy, Sungkyunkwan University, Suwon 16419, Korea. [17]These authors contributed equally: Mihee Oh, Seo Young Jang, Ji-Yoon Lee. ✉e-mail: bshan@kribb.re.kr; lesach@kribb.re.kr; gshwang@kbsi.re.kr; ewlee@kribb.re.kr

