## [Peer Review File · Nature Communications]

REVIEWER COMMENTS

Reviewer #1 (Remarks to the Author):

The manuscript by Oh et al describes the role of Lp PLA2 in ferroptosis resistance. Its inhibition by darapladib leads to the sensitization of cancer cells to ferroptosis. Author suggests in the title of the manuscript that darapladib induced sensitization is mediated via remodelling of lipid metabolism. Presented results might be of significant interest in exploration of cancer cell vulnerability towards ferroptotic cell death and its possible therapeutic exploitation. However, several important points needs to be addressed before the manuscript can be possibly accepted for the publication.

Points to address:

How specific darapladib towards Lp PLA2 inhibition?

Would inhibition of other PLA2 isoforms provide similar/stronger/weaker ferroptosis sensitization effects?

The part on the ferroptosis resistance in starvation is not clearly developed and need further investigation. How does it work? Via mTOR activation? Shift to MUFA? This point needs to be clarified.

Overall, in the manuscript different concentrations of both RSP3 and darapladib, different treatment times, cell densities are used. It is hard to follow and compare the results. Please provide the details in each figure legend.

Page 4, line 76 – correct linoleic acid as 18:2, not 18:3

Not sure that the statement that AA is synthesised in liver and then distributed to whole body is correct, many other cell types synthesise AA, not only cancer cells. Please correct or support the statement with the references to relevant literature

Page 6, line 109-110 - Show results on "validation with ferrostatin-1"

Figure 1A: please provide the whole dataset for the library screening as a supplementary dataset; what kind of library was used? Especially, did it include inhibitors for any other PLA2 isoforms?

Figure 1B and C – which treatment time was used? Cell density? Mention on the figure legend.

1D – what is high and what is low density? Mention on the figure legend. Which density was used for all further experiments? Why?

What are the different panels on figure 1F? what is the difference between 1st and 3rd and 2nd and 4th?

Figure S1B – show effect of darapladib alone (expected to be the same 60 % of cell viability as RSL3)

Figure 2B – here effect of darapladib alone gives around 85% of cell viability for Hs746T and 100% or 80% for SNU-484 cells? Why is that different in comparison with Figure S1A? which concentration of darapladib is used here now? And why is it so different for SNU-484 treated with darapladib only on the left and right panel? Again what was a cell density?

What is low RSL3 concentration (text to the figure 2D and E)? provide numbers!

Figure 2F upper panel x-axis states days, in the manuscript text those are hours

Figure 2D, E and F – BODIPY-C11 does not provide Lipid ROS (%), what is measured is oxidation of the dye itself – please correct for accuracy!

Figure 3E – GPX4 does look decreased to me on the blots upon darapladib treatment... please provide Source Data/ raw image files. Authors state "we observe no significant differences in protein

expression" which does not seem to be right for a number of proteins shown on the immunoblot Figure 3E – ELOVL5, LPCAT3 (both decreasing), FSP1 (increasing) and GPX4. Please provide raw data in replicates for reanalysis.

Page 8, line 171 – how exactly lysoPC include ROS production? Please clarify and support by the reference.

Page 8, line 174 – please comment on why darapladib failed the phase III clinical trials.

Page 8, lines 169-180 - this line of thoughts looks rather superficial and the conclusion "these data imply that the production of lysoPC by Lp-PLA2 might not be associated with ferroptosis" (which is probably true) is not supported by the data.

Lipidomics data in the way they are presented (lipids reported at species but not molecular species level) does not allow to make assumptions about the fatty acyl compositions of the different phospholipids.

Lipidomics experiments should be performed with darapladib in combination with RSL3 treatment as well so judge the role of the Lp PLA2 inhibition on ferroptosis. Comparison of DMSO vs darapladib doesn't tell us anything about the ferroptosis related remodelling of the lipidome. According to the data shown on Figure S1A 2 uM darapladib does not alter cell viability. Thus I am wondering how these lipidomics results can be connected with ferroptosis. Presented results rather reflect lipidome alterations in the presence of PLA2 inhibition and thus might indicate preferential substrates for the enzyme. Please present results accordingly. Based on the presented results PS lipids for instance might be considered as Lp PLA2 substrates, as well as PI.

Lipid IDs on the Figure 4A are hard to read, please increase the resolution of the labels.

Figure 4B – what kind of units are presented on the y- axis? how they were derived?

Figure 4 and corresponding text - Time component of the experiment is not discussed

Figure 6E – how the concentrations were measured? Include darapladib treatment alone here.

Please check figure S6 and S7 references in the text, doesn't fit to the content of the figures.

Cell viability was measured as ATP levels. Those would reflect a complex set of outcomes including mitochondrial (dys)functions and overall energy status of the cells. It is hard to directly relate it to the cell viability decreased due to the ferroptosis. Other measures (e.g. LDH release, any other markers relevant to plasma membrane rupture, pore formations, e.g. SytoxGreen or similar) would be much more informative to report ferroptotic (necrotic) cell death modality.

Lipidomic analysis – provide raw data using available repositories. How normalization was performed?

Cell count after collections of the cells? Protein concentrations?

Internal standards need to be added BEFORE not after the lipid extraction.

Identification strategy is quite unclear... identification based on the HMDB, MELIN and Lipid Maps implies identification on MS1 level which is not sufficient. How exactly MS/MS patterns were confirmed? which software was used? Please provide ID results table with m/z, RT, and identified fragments.

For MRM measurements, provide the list of transitions, normalization strategy, used internal standards, amounts of spiked internal standards, tables reporting measured peak area.

For oxPE and PE analysis, standards need to be added before the extraction, not after. Provide the list of transitions, normalization strategy, used internal standards, tables reporting measured peak area. oxPE can not be quantified using PE standard, as these lipids shown different polarities, ionization intensities, in-source fragmentation etc. Calibration curves are mentioned but not explained, all raw data and processed quantification data have to be provided with the manuscript.

Reviewer #2 (Remarks to the Author):

In this study, Oh et al. introduce PLA2G7 (aka, Lp-PLA2) as a novel regulator of cellular sensitivity to lipid peroxidation. Starting from a pharmacological screen for small molecules able to sensitise cells to ferroptosis the authors report that Darapladib, a known inhibitor of PLA2G7, sensitizes cells to cell death induced by GPX4 inhibitors. The authors propose that PLA2G7 acts intracellularly by a mechanism which the authors propose to be mediated by lipidomic changes. Following these observations, the authors provide proof of concept in xenografts that this combination can suppress tumour growth in vivo. The findings are generally interesting; the combination of a drug that can promote ferroptosis in vivo is certainly of importance. Nevertheless, some of these excitement is put of due to lack of data supporting the mechanism proposed here. My major criticism rests on the fact that it doesn't seem that sensitization effect conferred by Darapladib acts via its reported target. Therefore in its present form I believe the study is still in an immature stage for publication in Nature Communications.

- The data supporting that Darapladib is indeed acting via PLA2G7 is not convincing. While the combination of the inhibitor seems to generate an overall robust sensitization to GPX4 inhibitors, the genetic data is much less convincing. For example, if one compares the experiments in Fig S2B the IC50 for RSL3 in the presence of Darapladib drops from 10 μ M (derived from Figure 5B) to approximately 100nM whilst the IC50 of RSL3 in the PLA2G7 knockout is roughly 6 μ M. I believe this observation on its own would discard PLA2G7 as the target responsible for the sensitisation induced by Darapladib treatment. Moreover, the marginal effect observed with the overexpression of PLA2G7 in the sensitive cell lines does not make a strong case for the proposed role of PLA2G7 in of it being the target of Darapladib in this context.

- I believe the authors should consider developing further on the target identification – CRISPR dropout screen could be a powerful tool to narrow down the target of Darapladib in these cell lines. Though this would required a considerable amount of time.

- It is not explicitly stated how the lipoprotein-deficient serum was generated. In case this was purchased, I am afraid that the comparisons made through the study with the standard serum are not possible. Given that the authors state "lipoprotein deficiency generally slows the ferroptosis response", lipoprotein supplementation to LPDS would be a more meaningful comparison as this would be the only variable in the experiment.

Reviewer #3 (Remarks to the Author):

In this manuscript, the authors showed darapladib (Dara), an inhibitor of Lp-PLA2, sensitized the cancer cell to ferroptosis. Based on the findings, they found Lp-PLA2 as a negative regulator of ferroptosis. The ferroptosis sensitizing effect of Dara in combination with a GPX4 inhibitor was also examined in a xenograft animal study. Overall, the study was well controlled and performed, and the findings are interesting in the research field. However, several issues were raised to conclude the study, in particular the used in vivo model and the specificity of the drug.

1 The result of in vivo data is not sufficient to show the ferroptosis-sensitizing effect of Dara. The authors used PACMA31, a GPX4 inhibitor, in vivo. However, so far there is no established GPX4 inhibitor available for in vivo conditions. In previous reports, PACMA31 was used in vivo setting, but its ferroptosis-inducing effect has not yet been established. To demonstrate the ferroptosis-inducing effect of PACMA31, an additional series of data is required, such as checking the rescue effect by the treatment with in vivo available ferroptosis inhibitors, confirmation of (oxy)lipid peroxidation in the cancers utilizing high-resolution lipidomics, and exclusion of other types of cell death.

2. Due to the limitation of using GPX4 inhibitor in vivo, to support the in vivo effect of Dara, a xenograft model using GPX4 KO (or KD) cancer cell lines is required by using the cell lines that can proliferate without or with less expression of GPX4.

3. To confirm the mechanism of the ferroptosis sensitizing effect of Dara in vivo, lipidomics analysis would be necessary to check if the lipid profile is changed by the treatment as like in the in vitro condition.
- 4 As the authors mentioned, other PLA2 family members have also been reported as a negative regulators of ferroptosis. To show the specificity of Dara against Lp-PLA2, the authors could evaluate the ferroptosis sensitizing effect of Dara in the cells with si- or KO of other PLA2 genes related to ferroptosis sensitivity.
5. In Fig 5I, please show the change in RSL3 sensitivity by using multiple doses of RSL3. It is important data to demonstrate the pharmacological target of Dara is actually lp-PLA2.
6. As in comment #5. In Fig 5K, please show the data using multiple doses of RSL3.
7. The authors only focused on the role of Lp-PLA2 and Dara in cancer cells. However, please examine whether Dara shows the ferroptosis sensitizing effect in non-cancer cell lines. The expression level of Lp-PLA2 in non-cancer cell lines also could be compared to that in cancer cells.
8. The synergistic effect of Dara with other classes of ferroptosis inducers other than GPX4 inhibitors should be checked. In the study, they used only GPX4 inhibitors and Cys depletion for induction of ferroptosis.
9. Please show the protective effect of Fer1 with a cell viability curve (Fig 1F, s2B) to show whether Fer-1 completely rescued the synergistic effect of RSL3 with Dara.
9. In Fig 3D, it is well known that nutrients such as Vit E, selenium and CoQ10 contained in serum can suppress ferroptosis.
10. In Fig S4C, it is hard to compare the level of ACC. Please repeat the WB and quantify the band.
11. In S5A, as Lp-PLA2 is not a gene name, siLp-PLA2 looks strange. Please show the efficiency of siPLA2G7 and PLA2G KO by WB if possible.

Reviewer #1 (Remarks to the Author):

The manuscript by Oh et al describes the role of Lp PLA2 in ferroptosis resistance. Its inhibition by darapladib leads to the sensitization of cancer cells to ferroptosis. Author suggests in the title of the manuscript that darapladib induced sensitization is mediated via remodelling of lipid metabolism. Presented results might be of significant interest in exploration of cancer cell vulnerability towards ferroptotic cell death and its possible therapeutic exploitation. However, several important points needs to be addressed before the manuscript can be possibly accepted for the publication.

Response: We greatly appreciate the valuable comments from the reviewer, which improved our manuscript.

Points to address:

How specific darapladib towards Lp PLA2 inhibition?

Response: This comment raises the most important and critical question of our study. Darapladib is a highly selective Lp-PLA2 inhibitor with an in vitro IC50 of 0.25 nM, as revealed using recombinant Lp-PLA2 and LDL¹. Because of the significant association of Lp-PLA2 with cardiovascular events, two large phase III clinical trials of darapladib for coronary heart diseases have been conducted, but both trials failed^{2,3}.

In the revised version, we show that the CRISPR/Cas9-mediated deletion of *PLA2G7*, which encodes Lp-PLA2, sensitises cells to ferroptosis and leads to the accumulation of PE species containing PUFAs in a similar manner to darapladib treatment (Figs. 4 to 6). Importantly, darapladib did not sensitise *PLA2G7* KO cells to ferroptosis (Fig. 4k), suggesting that darapladib enhances ferroptosis in an Lp-PLA2-dependent manner. Finally, we demonstrate that darapladib effectively prevents the release of C20:4-d11 from PE-18:0/20:4-d11 (Fig. 6c)

Would inhibition of other PLA2 isoforms provide similar/stronger/weaker ferroptosis sensitization effects?

Response: This comment raises an important question, which was also raised by reviewer #3, because other PLA2 family members might affect ferroptosis due to their ability to cleave PUFAs such as arachidonic acid at the *sn*-2 position. Indeed, iPLA2 β is known to inhibit ferroptosis by cleaving oxidised arachidonic acid-containing PE (oxPE-18:0/20:4)⁴. Therefore, we tested the effect of other PLA2 isoforms on ferroptosis using pools of siRNAs against *PLA2G2A*, *PLA2G4A*, *PLA2G6*, and *PLA2G7*, which encode sPLA2, cPLA2, iPLA2, and Lp-PLA2, respectively. Interestingly, depletion of each PLA2 isoform, with the exception of *PLA2G4A*, induces RSL3-induced ferroptosis in SNU-484 and H1299 cells, suggesting that various PLA2 isoforms may play a crucial role in ferroptosis, although the detailed mechanism is unknown (Supplementary Fig. 16). To further confirm the involvement of the PLA2 isoform in ferroptosis, we employed several inhibitors of PLA2, such as (S)-BEL (an iPLA2 inhibitor), varespladib (an sPLA2 inhibitor also tested in phase III clinical trials), MAFP (a cPLA2/sPLA2 inhibitor), and MJ33 (an inhibitor of the PLA2 activity of PRDX6). Surprisingly, all PLA2 inhibitors with the exception of darapladib, had no effect on ferroptosis in Hs746T cells (Supplementary Fig. 1 and Fig. 16b). Interestingly, (S)-BEL itself was very toxic to H9c2 cells but accelerated ferroptosis, suggesting the critical role of iPLA2 in cardiomyocytes (Supplementary Fig. 16c). In addition, while darapladib completely prevented the release of arachidonic acid-d11 (AA-d11) from PE-18:0/20:4-d11 at 10 μ M, (S)-BEL showed little effect on PE

cleavage (Supplementary Fig. 6c). Therefore, we hypothesize that Lp-PLA2 is probably the most potent enzyme in maintaining the amount of arachidonic acid-containing PE in the cell and is therefore a crucial enzyme for ferroptosis resistance. We describe the impact of other PLA2 enzymes on ferroptosis in the manuscript as follows:

“other PLA2 isoforms may control ferroptosis by regulating PUFA-containing PL abundance. Although the depletion of several PLA2 isoforms resulted in context-dependent promotion of ferroptosis, the inhibition of each PLA2 isoform with their inhibitors had no obvious effect on ferroptosis in cancer cells (Supplementary Fig. 16a, b). Interestingly, (S)-BEL, an inhibitor of iPLA2, is itself toxic to H9c2 cells at high concentrations but slightly enhances ferroptosis, possibly due to its ability to inhibit iPLA2 activities against oxidised and non-oxidised SAPE (Supplementary Fig. 16c)^{5,6}. Notably, darapladib sensitised various cell lines, including H9c2 cells, to ferroptosis, suggesting that Lp-PLA2 may be a common regulator of ferroptosis that conserves PEs containing PUFAs.” (Page 17, lines 4-11)

The part on the ferroptosis resistance in starvation is not clearly developed and need further investigation. How does it work? Via mTOR activation? Shift to MUFA? This point needs to be clarified.

Response: As noted by the reviewer, starvation-induced mTOR activation could affect ferroptosis because mTOR is associated with ferroptosis⁷. To test this possibility, we determined the levels of phospho-S6K to monitor mTOR activity, but the levels of phospho-S6K were unaffected by lipoprotein deficiency (Supplementary Fig. 5a). It is also possible that lipoprotein deficiency induces lipid reprogramming to decrease PUFAs and increase MUFAs, but confirmation of this possibility requires further lipidomics analysis. As suggested by reviewer 2, we tested whether supplementation with lipoprotein can rescue ferroptosis sensitivity and found that supplementation with HDL, but not LDL and VLDL, resensitised cells to ferroptosis under lipoprotein deficiency, suggesting that HDL might contribute to ferroptosis, although the underlying mechanism is currently unclear (Supplementary Fig. 5b). These results are discussed on page 8, lines 9-13:

“Although starvation stress may activate the mTOR pathway, which can suppress ferroptosis, the levels of phospho-S6K were unaffected by lipoprotein deficiency (Supplementary Fig. 5a)^{8,9}. Interestingly, supplementation with HDL, but not LDL or VLDL, resensitised cells to ferroptosis under lipoprotein deficiency, suggesting that HDL may contribute to ferroptosis, although the underlying mechanism is unclear (Supplementary Fig. 5b)”

Overall, in the manuscript different concentrations of both RSL3 and darapladib, different treatment times, cell densities are used. It is hard to follow and compare the results. Please provide the details in each figure legend.

Response: We apologise for any inconvenience caused by lack of detailed information. We now provide detailed information in each figure legend.

Page 4, line 76 – correct linoleic acid as 18:2, not 18:3

Response: Thank you for catching our mistake, and we have corrected this error on page 4, line 24.

Not sure that the statement that AA is synthesised in liver and then distributed to whole body is correct, many

other cell types synthesise AA, not only cancer cells. Please correct or support the statement with the references to relevant literature

Response: Thank you for indicating our misinformation. We have removed the inaccurate sentence as suggested on page 4, line 26, to page 5, line 1.

“AA can be synthesised from the n-6 essential fatty acid linoleic acid (LA, 18:2), the most abundant fatty acid in serum and plasma. Our recent study suggests that certain gastric cancer cells depend on this PUFA synthesis pathway and thus show hypersensitivity to ferroptosis.”

Page 6, line 109-110 - Show results on “validation with ferrostatin-1”

Response: We have added the results from our ferrostatin-1 validation for the candidate drugs in Supplementary Fig. 1b.

Figure 1A: please provide the whole dataset for the library screening as a supplementary dataset; what kind of library was used? Especially, did it include inhibitors for any other PLA2 isoforms?

Response: We have provided the whole dataset for library screening in Supplementary Table 1 and described this dataset in the Methods section. The library we used is a metabolism compound library purchased from SelleckChem (L3700) and contains only varespladib, an sPLA2 inhibitor, as noted in Fig. 1a. In revised Supplementary Fig. 1a, we provide the results from all the compounds tested as a heatmap.

Figure 1B and C – which treatment time was used? Cell density? Mention on the figure legend.

Response: We now describe the treatment time and the number of cells seeded in the legend of Fig. 1b, c as follows:

(b) Relative viability of Hs746T and SNU-484 cells treated with increasing concentrations of RSL3 and/or 2 μ M darapladib for 20 h. Cells were plated at 30,000 Hs746T cells/well and 40,000 SNU-484 cells/well in 200 μ L of media. (c) Crystal violet staining of cells treated with RSL3 and/or 2 μ M darapladib for 48 h. Cells were plated at 20,000 Hs746T cells/well and 25,000 SNU-484 cells/well in 200 μ L of media.

1D – what is high and what is low density? Mention on the figure legend. Which density was used for all further experiments? Why?

Response: We have added the number of cells used for high and low density in Fig. 1d as follows, and all further experiments were performed at high density. We believe that the induction of ferroptosis at high density would be more meaningful because many cancer cells are attached to each other in tumour tissue.

(e) Relative viability of cells at high (30,000 Hs746T cells/well and 40,000 SNU-484 cells/well) and low (20,000 Hs746T cells/well and 25,000 SNU-484 cells/well) densities upon RSL3 and 2 μ M darapladib treatment.

What are the different panels on figure 1F? what is the difference between 1st and 3rd and 2nd and 4th?

Response: We apologise for the inaccurate notation. We have now indicated the name of the cell line in each panel.

Figure S1B – show effect of darapladib alone (expected to be the same 60 % of cell viability as RSL3)

Response: We repeated the experiment with darapladib alone and found that 2 μ M darapladib alone had little effect on cell death measured by PI uptake (Fig 1e) but decreased cell viability by approximately less than 20% (Figs. 2b, 3b, d, Supplementary Fig. 2b), suggesting that darapladib may influence cell proliferation rather than cell death. Furthermore, we found that darapladib becomes more toxic when the cell density is low, which may lead to an experimental error in the assessment of darapladib toxicity (Supplementary Fig. 2d).

Figure 2B – here effect of darapladib alone gives around 85% of cell viability for Hs746T and 100% or 80% for SNU-484 cells? Why is that different in comparison with Figure S1A? which concentration of darapladib is used here now? And why is it so different for SNU-484 treated with darapladib only on the left and right panel? Again what was a cell density?

Response: We apologise again for any confusion caused by this variance and the lack of detailed experimental information. The toxicity induced by darapladib alone may vary between experimental conditions, particularly the cell density, because darapladib becomes more toxic when the cell density is low (Supplementary Fig. 2d). In all experiments, 2 μ M darapladib was used at a normal density as described in the figure legends, and darapladib alone decreased cell viability by less than 20%, possibly due to experimental errors (Figs. 2b, 3b, d, Supplementary Fig. 2b). Nevertheless, darapladib and RSL3 treatment greatly induced ferroptosis, suggesting that darapladib indeed promotes RSL3-induced cell death.

What is low RSL3 concentration (text to the figure 2D and E)? provide numbers!

Response: Thank you for pointing out the unclear information, and we now provide the exact concentration of RSL3 (0.1 μ M) used in our experiments.

Figure 2F upper panel x-axis states days, in the manuscript text those are hours

Response: Thank you for pointing out the error. We have corrected the notation of the x-axis to hours.

Figure 2D, E and F – BODIPY-C11 does not provide Lipid ROS (%), what is measured is oxidation of the dye itself – please correct for accuracy!

Response: Thank you for your suggestion. We have labelled the % of oxidised C11 BODIPY in the figure and described it as the % of cells with oxidised C11 BODIPY in the figure legend.

Figure 3E – GPX4 does look decreased to me on the blots upon darapladib treatment... please provide Source Data/ raw image files. Authors state “we observe no significant differences in protein expression” which does not seem to be right for a number of proteins show on the immunoblot Figure 3E – ELOVL5, LPCAT3 (both decreasing), FSP1 (increasing) and GPX4. Please provide raw data in replicates for reanalysis.

Response: We agree with the reviewer that there are some fluctuations in the protein levels in the original Figure 3E. Considering that GPX4 is an abundant protein, such a small change may not affect ferroptosis sensitivity.

Nevertheless, we repeated the same experiment and found no meaningful changes in these protein levels. In addition, *PLA2G7* KO cells also express similar levels of ferroptosis-related proteins, although NRF2 and FTH1 are found at slightly higher levels (these proteins are more likely to inhibit rather than promote ferroptosis). Therefore, we did not provide the statistical data but corrected the sentence as follows:

“we observed no significant differences in protein expression upon RSL3 and/or darapladib treatment” (Page 8, lines 27-28).

Page 8, line 171 – how exactly lysoPC include ROS production? Please clarify and support by the reference.

Response: We apologise for the missing references on the association of lysoPC with ROS and atherosclerosis. It has long been recognised that lysoPC induces ROS, and various mechanisms have been proposed, including NADPH oxidase and NOX^{10, 11, 12}. We have added these references to the manuscript on page 9, line 6.

Page 8, line 174 – please comment on why darapladib failed the phase III clinical trials.

Response: Darapladib failed to meet the primary endpoint of reducing the risk of major coronary events in two large phase III clinical trials, STABILITY and SOLID-TIMI, raising questions about the role of Lp-PLA2 as a causal factor or an early checkpoint^{2, 3}. We modified the sentence as follows:

“although darapladib recently failed in phase III clinical trials due to lack of efficacy” (Page 9, lines 7-8)

Page 8, lines 169-180 - this line of thoughts looks rather superficial and the conclusion “these data imply that the production of lysoPC by Lp-PLA2 might not be associated with ferroptosis” (which is probably true) is not supported by the data.

Response: As the reviewer noted, we revised the sentence as follows:

“Although we cannot rule out the possibility that Lp-PLA2-mediated lysoPC production remains linked to ferroptosis, our findings point to the existence of another dominant mechanism through which Lp-PLA2 regulates ferroptosis.” (Page 9, lines 14-16)

Lipidomics data in the way they are presented (lipids reported at species but not molecular species level) does not allow to make assumptions about the fatty acyl compositions of the different phospholipids.

Response: We have provided the specific fatty acyl chains for each lipid.

Lipidomics experiments should be performed with darapladib in combination with RSL3 treatment as well so judge the role of the Lp PLA2 inhibition on ferroptosis. Comparison of DMSO vs darapladib doesn't tell us anything about the ferroptosis related remodelling of the lipidome. According to the data shown on Figure S1A 2 uM daraplabid does not alter cell viability. Thus I am wondering how these lipidomics results can be connected with ferroptosis. Presented results rather reflect lipidome alterations in the presence of PLA2 inhibition and thus might indicate preferential substrates for the enzyme. Please present results accordingly.

Response: We completely agree with this comment that lipidomic changes during the ferroptotic process can provide a critical clue on the mechanism of darapladib in ferroptosis sensitisation. Several studies suggest that during ferroptosis, the levels of several phospholipids with PUFAs appear to decrease due to oxidative damage, which makes it difficult to interpret the increase in phospholipids upon Lp-PLA2 inhibition. In this study, we hypothesize that a “pro-ferroptotic lipid state” may be induced by the inhibition of Lp-PLA2, which renders cells vulnerable to ferroptosis. Although cotreatment with darapladib and RSL3 also sensitises cells to ferroptosis, darapladib-induced lipidomic changes could contribute to ferroptosis because we observed marked changes within 1 hour of darapladib treatment.

Based on the presented results PS lipids for instance might be considered Lp PLA2 substrates, as well as PI.

Response: Because we agree with this comment, we added the following sentence.

“In addition, other phospholipids, such as PI, PS, and PG, may be the target of Lp-PLA2 because these PLs and lysoPLs are oppositely regulated by Lp-PLA2 deficiency or inhibition (Figs. 5, 6 and Supplementary Figs. 9, 10).”
(Page 11, lines 12-14)

Lipid IDs on the Figure 4A are hard to read, please increase the resolution of the labels.

Response: We apologise for any inconvenience caused by the low-quality figures during the initial submission process. We have now provided increased the resolution of the labels.

Figure 4B – what kind of units are presented on the y- axis? how they were derived?

Response: We summed the intensities by major lipid classes and divided the sum of lipid classes by the detected total lipid intensities. Therefore, the y-axis indicates the percentage of the lipid class. We added axis descriptions to the legends of Supplementary Figs. 9 and 10.

“Supplementary Fig. 9. PLA2G7 deficiency results in the accumulation of PE and PE-p species and the depletion of lysoPE species.

(a) Proportions of various lipid classes normalised by the total lipids detected by LC–MS/MS in WT and PLA2G7 KO H1299 cells. The y-axis indicates the percentage of the lipid class.”

“Supplementary Fig. 10. Darapladib induces the accumulation of PE and PE-p species and decreases in lysoPE and MUFA species.

(a) Proportions of various lipid classes normalised by the total lipids detected by LC–MS/MS in Hs746T cells treated with darapladib. The y-axis indicates the percentage of the lipid class.”

Figure 4 and corresponding text - Time component of the experiment is not discussed

Response: First, we revised the legend of the revised Fig. 6a by adding time information to the Results section, and we added the following sentences:

“These lipidomic changes were observed as early as 1 h and lasted for 4 h, indicating that phospholipid remodelling, known as the Lands cycle, occurs very rapidly within the cell.” (Page 11, lines 8-9)

Figure 6E – how the concentrations were measured? Include darapladib treatment alone here.

Response: We conducted quantitative analyses of the intracellular PE-18:0/20:4+2O, PE-18:0/20:4 and PE-18:0/22:4 concentrations using calibration curves obtained with PE-18:0/20:4+2O and PE-18:0/20:4 standards, respectively. The concentrations were normalised to the protein levels, and PE/15:0/18:1-d7 was used as an internal standard. We added a detailed description of the quantification process to the Methods section. In addition, the measurement information, such as MRM transitions, cone voltage, collision energy, retention time, and calibration curve equation, including type, weighting, and dynamic range, is summarised in Supplementary Table 2. We repeated this experiment with darapladib alone and found that darapladib alone only marginally increased the PE-18:0/20:4+2O levels, possibly due to the increase in total PE-18:0/20:4, and this information is now presented in the revised Supplementary Fig. 11a. The y-axis indicates the response per protein.

“The individual concentrations, including those of PE-18:0/20:4+2O, PE-18:0/20:4 and PE-18:0/22:4, were normalised to the protein levels.” (Page 24, lines 28-29)

“Oxidised PE and PE-18:0/20:4 standards were used for optimisation of the conditions for each metabolite and for preparation of a series of calibration solutions to generate calibration curves. Details of the measurement are included in Supplementary Table 2.” (Page 25, lines 5-8)

Please check figure S6 and S7 references in the text, doesn't fit to the content of the figures.

Response: We thank the reviewer for pointing out this discrepancy. In the Discussion section, we corrected the original Fig. S6 to the revised Supplementary Fig. 15.

Cell viability was measured as ATP levels. Those would reflect a complex set of outcomes including mitochondrial (dys)functions and overall energy status of the cells. It is hard to directly relate it to the cell viability decreased due to the ferroptosis. Other measures (e.g. LDH release, any other markers relevant to plasma membrane rupture, pore formations, e.g. SytoxGreen or similar) would be much more informative to report ferroptotic (necrotic) cell death modality.

Response: Thank you for the important suggestion. Indeed, although darapladib reduced cell viability by less than 20%, it had only a small effect on cell death as measured by the PI uptake and released LDH levels (revised Fig. 1e and Supplementary Fig. 2). We also investigated the effect of *PLA2G7* deletion on ferroptosis via cell viability (ATP), PI uptake, and LDH release (Figs. 4 and 5).

Lipidomic analysis – provide raw data using available repositories. How normalization was performed? Cell count after collections of the cells? Protein concentrations?

Internal standards need to be added BEFORE not after the lipid extraction.

Identification strategy is quite unclear... identification based on the HMDB, MELIN and Lipid Maps implies identification on MS1 level which is not sufficient. How exactly MS/MS patterns were confirmed? which software was used? Please provide ID results table with m/z, RT, and identified fragments.

Response: All data were normalised to the detected total lipids. We added SPLASH Lipidomix and FFA 13:0 as internal standards to the extraction solution. To identify lipids in cell and tumour tissues, we identified the lipids on MS1 and MS2 by using various databases including LIPID Maps, HMDB and METLIN using PeakView (Sciex, Concord, ON, Canada) and MassLynx (Waters, Milford, MA, USA) software. We then confirmed the MS2 spectra and retention time using an in-house library acquired from commercial lipid standards. As the reviewer pointed out, we have provided tables with the ID results as well as m/z and retention time in Supplementary Data 1, 2, and 9. Additionally, we added representative MS2 spectra of tumour tissues among the lipidomics data along with lipid standard compounds in Supplementary Figure 17 instead of identified fragments. We revised the ID process in the Methods section.

“Tridecanoic acid (0.5 µM, FFA 13:0, Sigma–Aldrich, USA) and a 50-fold dilution of SPLASH Lipidomix (Avanti Polar Lipids, USA) were mixed with the extraction solution as internal standards.” (Page 23, lines 5-7)

“All data were normalised to the detected total lipids. Lipid metabolites were identified by comparing experimental data with online databases (DBs; HMDB, METLIN, and LIPID MAPS) and our in-house library. Identification was confirmed using the MS/MS patterns and retention times of lipid standard compounds. Representative MS/MS spectra of various lipid classes are presented in Supplementary Fig. 17.” (Page 23, lines 29 to Page 24, lines 1-4)

For MRM measurements, provide the list of transitions, normalization strategy, used internal standards, amounts of spiked internal standards, tables reporting measured peak area.

Response: Based on the reviewer’s comment, we added Supplementary Tables 1, 2, and 3 and Supplementary Data 3, 4, 5, 6, 7, and 8, including information related to the list of transitions and measured peak area. Additionally, we describe the internal standards used and the amounts of spiked internal standards in the Methods section.

“Tridecanoic acid (0.5 µM, FFA 13:0, Sigma–Aldrich, USA) and a 50-fold dilution of SPLASH Lipidomix (Avanti Polar Lipids, USA) were mixed with the extraction solution as internal standards.” (Page 23, lines 5-7)

“All data were normalised to the detected total lipids.” (Page 23, lines 29 to Page 24, lines 1)

“Subsequently, 0.5 µM FFA 13:0 and 500 ppb lysoPC 18:1-d7 (Avanti Polar Lipids, USA) were mixed with methanol:acetonitrile (50:50, v/v) solution, and the extraction process was the same as that used for lipidomic analysis. The intensities were normalised to the cell numbers.” (Page 24, lines 18-21)

“A total of 100 ppb PE-15:0/18:1-d7 (Sigma–Aldrich, USA) was mixed with methanol:water (80:20, v/v) or redissolution solvent as an internal standard. The individual concentrations, including those of PE-18:0/20:4+2O, PE-18:0/20:4 and PE-18:0/22:4, were normalised to the protein levels.” (Page 24, lines 26-29)

For oxPE and PE analysis, standards needs to be added before the extraction, not after. Provide the list of transitions, normalization strategy, used internal standards, tables reporting measured peak area. oxPE can not be quantified using PE standard, as these lipids shown different polarities, ionization intensities, in-source fragmentation etc. Calibration curves are mentioned but not explained, all raw data and processed quantification data have to be provided with the manuscript.

Response: As the reviewer pointed out, we added an internal standard in additional experiments before the extraction process. As mentioned above, we quantified the intracellular PE-18:0/20:4+2O concentrations using calibration curves from PE-18:0/20:4+2O standards, not just a PE standard. The individual concentrations, including those of PE-18:0/20:4+2O, PE-18:0/20:4 and PE-18:0/22:4, were normalised to the protein levels, and PE/15:0/18:1-d7 was used as an internal standard. We revised the detailed description of the quantification process in the Methods section and added Supplementary Table 2, which includes detailed information related to the analysis: transition, retention time and collision energy, calibration curve equation, type, weighting, and dynamic range. Additionally, we have provided the measured data in Supplementary Data 3, 6, and 7 according to the reviewer’s suggestion.

“The lipid extracts were dried under nitrogen gas and reconstituted in isopropanol/acetonitrile/water 50:25:25 (v/v/v). A total of 100 ppb PE-15:0/18:1-d7 (Sigma–Aldrich, USA) was mixed with methanol:water (80:20, v/v) or redissolution solvent as an internal standard. The individual concentrations, including those of PE-18:0/20:4+2O, PE-18:0/20:4 and PE-18:0/22:4, were normalised to the protein levels.” (Page 24, lines 25-29)

“Oxidised PE and PE-18:0/20:4 standards were used for optimisation of the conditions for each metabolite and for preparation of a series of calibration solutions to generate calibration curves.” (Page 25, lines 5-7)

Reviewer #2 (Remarks to the Author):

In this study, Oh et al. introduce PLA2G7 (aka, Lp-PLA2) as a novel regulator of cellular sensitivity to lipid peroxidation. Starting from a pharmacological screen for small molecules able to sensitise cells to ferroptosis the authors report that Darapladiib, a known inhibitor of PLA2G7, sensitizes cells to cell death induced by GPX4 inhibitors. The authors propose that PLA2G7 acts intracellularly by a mechanism which the authors propose to be mediated by lipidomic changes. Following these observations, the authors provide proof of concept in xenografts that this combination can suppress tumour growth in vivo. The findings are generally interesting; the combination of a drug that can promote ferroptosis in vivo is certainly of importance. Nevertheless, some of these excitement is put of due to lack of data supporting the mechanism proposed here. My major criticism rests on the

fact that it doesn't seem that sensitization effect conferred by Darapladib acts via its reported target. Therefore in its present form I believe the study is still in an immature stage for publication in Nature Communications.

Response: We appreciate the thorough review and constructive criticism of our work. We agree that our initial version lacked evidence, to some extent, of the mechanism by which darapladib sensitises ferroptosis via its known target, Lp-PLA2; thus, we attempted to provide evidence of this by including the following experiments.

1. We established H1299 cells in which *PLA2G7*, which encodes Lp-PLA2, was knocked down and observed increased sensitivity to ferroptosis in these cells comparable to that observed with darapladib treatment.
2. The lipid profiles of *PLA2G7* KO and WT H1299 cells were analysed, and *PLA2G7* KO cells exhibited lipidomic changes similar to those of cells treated with darapladib.
3. We also provide evidence showing that darapladib can inhibit PE-18:0/20:4 cleavages in vitro.
4. We also propose a docking simulation model for the interaction between Lp-PLA2 and several phospholipids such as PE-18:0/20:4 and PE-18:0/20:4-OOH, which suggests a potential activity of Lp-PLA2 towards various phospholipid species.

- The data supporting that Darapladib is indeed acting via PLA2G7 is not convincing. While the combination of the inhibitor seems to generate an overall robust sensitization to GPX4 inhibitors, the genetic data is much less convincing. For example, if one compares the experiments in Fig S2B the IC50 for RSL3 in the presence of Darapladib drops from 10µM (derived from Figure 5B) to approximately 100nM whilst the IC50 of RSL3 in the PLA2G7 knockout is roughly 6µM. I believe this observation on its own would discard PLA2G7 as the target responsible for the sensitisation induced by Darapladib treatment. Moreover, the marginal effect observed with the overexpression of PLA2G7 in the sensitive cell lines does not make a strong case for the proposed role of PLA2G7 in of it being the target of Darapladib in this context.

Response: Thank you for your precise and comprehensive point. We were also concerned about this issue because *PLA2G7*-deficient YCC16 cells are not very sensitive to ferroptosis compared with darapladib treatment, as noted by this reviewer. We suspect that *PLA2G7* WT and KO YCC16 cells generated using the CRISPR/Cas9 system may acquire overall ferroptosis resistance during single-cell cloning. We therefore re-established the *PLA2G7* WT/KO H1299 cell lines and found that the RSL3 sensitivity of the parental and cloned *PLA2G7* WT in H1299 cells was similar. Using these cells, we found that *PLA2G7* KO cells are hypersensitive to ferroptosis in response to RSL3 treatment or cysteine deficiency (revised Fig. 4a-j). Although H1299/*PLA2G7* KO cells were slightly less effective at promoting ferroptosis than darapladib, darapladib no longer promoted ferroptosis in these cells, suggesting that darapladib targets Lp-PLA2 (revised Fig. 4k). We then analysed the lipid profile of *PLA2G7* KO cells and found that darapladib treatment increased the levels of most PE species and decreased those of lysoPE in a similar manner (revised Fig. 5f, g), suggesting that darapladib accelerates ferroptosis by increasing pro-ferroptotic phospholipids through Lp-PLA2 inhibition.

To ascertain whether darapladib indeed has the ability to protect against the cleavage of PE species, we performed an in vitro cleavage assay using cell lysates and PE-18:0/20:4-d11 and then detected the cleaved product, AA (C20:4)-d11. In this assay, we found that while cell lysates efficiently cleaved PE-18:0/20:4, PE cleavage was blunted in the presence of darapladib (revised Fig. 6c). Since Lp-PLA2 is known to preferentially

cleave PC species containing oxidative truncated fatty acyl chains, we also tested whether Lp-PLA2 is responsible for the cleavage of PE using cell lysates from *PLA2G7* KO cells. As a result, lysates from *PLA2G7* KO cells had a reduced ability to cleave PE-18:0/20:4-d11, indicating that Lp-PLA2 contributes to the PE deacylation cycle (revised Supplementary Fig. 12c).

Finally, we proposed a docking simulation model for the interaction between Lp-PLA2 and various phospholipids species containing arachidonic acid, as well as oxidised arachidonic acid, which was suggested to be a target of Lp-PLA2¹³. The binding affinity values calculated suggested that Lp-PLA2 showed broad substrate specificity including PE-18:0/20:4 (Supplementary Fig. 12). Nevertheless, we noticed that the suggested binding affinity was relatively low, suggesting that there may be additional mechanism that determine the specificity of Lp-PLA2 in cells. Direct determination of the real tertiary structure of Lp-PLA2 and phospholipids in condition similar to the cell membrane should be investigated in the future.

- I believe the authors should consider developing further on the target identification – CRISPR dropout screen could be a powerful tool to narrow down the target of Darapladib in these cell lines. Though this would require a considerable amount of time.

Response: We appreciate the reviewer for the valuable comment. Although we still believe Lp-PLA2 is a major target in darapladib-mediated sensitisation to ferroptosis, we cannot rule out the possibility of other unknown targets being involved in ferroptosis regulation because the lipidomic changes induced by Lp-PLA2 deletion and inhibition are similar but not identical, and darapladib has a stronger effect on ferroptosis, as noted by this reviewer. In addition, the identification of new darapladib targets may shed light on the reasons for the failure of darapladib in a clinical trial for coronary artery disease. However, we believe that this is beyond the scope of this study and will be investigated in the future.

- It is not explicitly stated how the lipoprotein-deficient serum was generated. In case this was purchased, I am afraid that the comparisons made through the study with the standard serum are not possible. Given that the authors state “lipoprotein deficiency generally slows the ferroptosis response”, lipoprotein supplementation to LPDS would be a more meaningful comparison as this would be the only variable in the experiment.

Response: We appreciate the excellent suggestion to investigate the role of lipoprotein in ferroptosis. As this reviewer predicted, we used commercialised lipoprotein-deficient human serum (LPDS) from Sigma–Aldrich (S5519). We therefore investigated whether lipoprotein supplementation to LPDS can restore ferroptosis sensitivity and found that supplementation with HDL, but not LDL or VLDL, resensitises cells to ferroptosis under lipoprotein deficiency, suggesting that HDL may contribute to ferroptosis, although the underlying mechanism is unclear. These results are presented in Supplementary Fig. 5a and described as follows:

“Although starvation stress may activate the mTOR pathway, which can suppress ferroptosis, the levels of phospho-S6K were unaffected by lipoprotein deficiency (Supplementary Fig. 5a)^{8,9}. Interestingly, supplementation with HDL, but not LDL or VLDL, resensitised cells to ferroptosis under lipoprotein deficiency, suggesting that HDL may contribute to ferroptosis, although the underlying mechanism is unclear (Supplementary Fig. 5b). (Page 8, lines 9-13).

Reviewer #3 (Remarks to the Author):

In this manuscript, the authors showed darapladib (Dara), an inhibitor of Lp-PLA2, sensitized the cancer cell to ferroptosis. Based on the findings, they found Lp-PLA2 as a negative regulator of ferroptosis. The ferroptosis sensitizing effect of Dara in combination with a GPX4 inhibitor was also examined in a xenograft animal study. Overall, the study was well controlled and performed, and the findings are interesting in the research field. However, several issues were raised to conclude the study, in particular the used in vivo model and the specificity of the drug.

Response: We greatly appreciate the thorough review and positive and valuable comments, which improved our manuscript. In the revised version, we attempt to provide evidence of the specificity of darapladib, its role in vivo, and other issues raised, as shown below.

1 The result of in vivo data is not sufficient to show the ferroptosis-sensitizing effect of Dara. The authors used PACMA31, a GPX4 inhibitor, in vivo. However, so far there is no established GPX4 inhibitor available for in vivo conditions. In previous reports, PACMA31 was used in vivo setting, but its ferroptosis-inducing effect has not yet been established. To demonstrate the ferroptosis-inducing effect of PACMA31, an additional series of data is required, such as checking the rescue effect by the treatment with in vivo available ferroptosis inhibitors, confirmation of (oxy)lipid peroxidation in the cancers utilizing high-resolution lipidomics, and exclusion of other types of cell death.

Response: We appreciate the valuable comment from the reviewer. In a previous study, PACMA31 was pulled down with GPX4, and the covalent binding of PACMA31 to GPX4 was further confirmed by LC-MS/MS¹⁴. We also found that PACMA31 causes a GPX4 band shift, implying a covalent interaction with GPX4 in cells (revised Fig. 7a). Furthermore, a cellular thermal shift assay (CETSA) with cell lysates showed increased thermal stability of GPX4 in the presence of PACMA31, implying direct binding (revised Fig. 7b)¹⁵. Although the previous study used 10 μ M PACMA31, PACMA31 is highly sensitive to cancer cells with IC₅₀ values of less than 1 μ M, and this effect was completely rescued by ferrostatin-1 (Fig. 7c and Supplementary Fig. 14a). These findings strongly suggest that PACMA31 is a GPX4 inhibitor that induces ferroptosis in cells.

We then investigated the ferroptosis-inducing effect of PACMA31 in a mouse xenograft model and discovered that although PACMA31 had no significant effect on tumour growth in our new experiment, mice treated with PACMA31 and darapladib showed substantial antitumour activity (revised Fig. 7g, h). Interestingly, ferrostatin-1 treatment reversed the PACMA31/darapladib-induced tumour retardation, implying that ferroptosis is involved in tumour suppression (revised Fig. 7g, h). Furthermore, our analysis of the global lipid profile of xenografted tumours revealed that tumours treated with darapladib (darapladib alone, darapladib + PACMA31, and darapladib + PACMA31 + Fer-1) had a substantial increase in PE species, including PE-18:0/20:4 (revised Supplementary Fig. 14). However, these changes were not dramatic compared to those in cells, possibly due to the contamination of noncancer cells in tumour tissues. In addition, because specific oxygenated phospholipids have been proposed as cell death markers^{16,17}, it is worthwhile to investigate oxPLs in tumour tissue following treatment. However, due to time and technical constraints, the detection of each oxidised phospholipid was not possible in this study.

2. Due to the limitation of using GPX4 inhibitor in vivo, to support the in vivo effect of Dara, a xenograft model using GPX4 KO (or KD) cancer cell lines is required by using the cell lines that can proliferate without or with less expression of GPX4.

Response: To validate the in vivo effect of darapladib, we established H1299 cell lines stably expressing lentiviral shRNA for GPX4, which was selected in the presence of Fer-1 to prevent ferroptosis upon GPX4 depletion. Upon removal of ferrostatin-1, GPX4-depleted cells underwent ferroptosis with an increase in lipid peroxidation, which was further enhanced by darapladib, confirming that darapladib sensitises GPX4 depletion-induced ferroptosis (Supplementary Fig. 4). We also attempted to use these cells in a xenograft study, but mice injected with GPX4-depleted cells exhibited severe defects in tumour growth; thus, it was difficult to evaluate the effect of darapladib on these mouse models.

3. To confirm the mechanism of the ferroptosis sensitizing effect of Dara in vivo, lipidomics analysis would be necessary to check if the lipid profile is changed by the treatment as like in the in vitro condition.

Response: As indicated in our response to comment #1 above, we observed a substantial increase in PE species, including PE-18:0/20:4.

4 As the authors mentioned, other PLA2 family members have also been reported as a negative regulators of ferroptosis. To show the specificity of Dara against Lp-PLA2, the authors could evaluate the ferroptosis sensitizing effect of Dara in the cells with si- or KO of other PLA2 genes related to ferroptosis sensitivity.

Response: Thank you for this important question, which was also raised by reviewer #3, because other PLA2 family members possibly affect ferroptosis due to their ability to cleave PUFAs, such as arachidonic acid, at the sn-2 position. Indeed, iPLA2 β is known to inhibit ferroptosis by cleaving oxidised arachidonic acid-containing PE (oxPE-18:0/20:4)⁴. Therefore, we tested the effect of other PLA2 isoforms on ferroptosis using pools of siRNAs against *PLA2G2A*, *PLA2G4A*, *PLA2G6*, and *PLA2G7*, which encode sPLA2, cPLA2, iPLA2, and Lp-PLA2, respectively. Interestingly, depletion of each PLA2 isoform, with the exception of *PLA2G4A*, induces RSL3-induced ferroptosis in SNU-484 and H1299 cells, suggesting that various PLA2 isoforms may play a crucial role in ferroptosis, although the detailed mechanism is unknown (Supplementary Fig. 16). To further confirm the involvement of the PLA2 isoform in ferroptosis, we employed several inhibitors of PLA2, such as (S)-BEL (an iPLA2 inhibitor), varespladib (an sPLA2 inhibitor also tested in phase III clinical trials), MAFP (a cPLA2/sPLA2 inhibitor), and MJ33 (an inhibitor of the PLA2 activity of PRDX6). Surprisingly, all PLA2 inhibitors, with the exception of darapladib, had no effect on ferroptosis in Hs746T cells (Supplementary Fig. 1 and Fig. 16b). Interestingly, (S)-BEL itself was very toxic to H9c2 cells but accelerated ferroptosis, suggesting the critical role of iPLA2 in cardiomyocytes (Supplementary Fig. 16c). In addition, while darapladib completely prevented the release of arachidonic acid-d11 (AA-d11) from PE-18:0/20:4-d11 at 10 μ M, (S)-BEL showed little effect on PE cleavage (Supplementary Fig. 6c). Therefore, we hypothesise that Lp-PLA2 is probably the most potent compound in maintaining the amount of arachidonic acid-containing PE in the cell and is therefore a crucial enzyme for ferroptosis resistance. We describe the impact of other PLA2 enzymes on ferroptosis in the manuscript as follows:

“other PLA2 isoforms may control ferroptosis by regulating PUFA-containing PL abundance. Although the depletion of several PLA2 isoforms resulted in context-dependent promotion of ferroptosis, the inhibition of each PLA2 isoform with their inhibitors had no obvious effect on ferroptosis in cancer cells (Supplementary Fig. 16a, b). Interestingly, (S)-BEL, an inhibitor of iPLA2, is itself toxic to H9c2 cells at high concentrations but slightly enhances ferroptosis, possibly due to its ability to inhibit iPLA2 activities against oxidised and non-oxidised SAPE (Supplementary Fig. 16c)^{5,6}. Notably, darapladib sensitised various cell lines, including H9c2 cells, to ferroptosis, suggesting that Lp-PLA2 may be a common regulator of ferroptosis that conserves PEs containing PUFAs (Supplementary Fig. 16).” (Page 17, lines 4-11)

5. In Fig 5I, please show the change in RSL3 sensitivity by using multiple doses of RSL3. It is important data to demonstrate the pharmacological target of Dara is actually lp-PLA2.

Response: This comment describes the importance of supporting the on-target effect of darapladib on Lp-PLA2. As also pointed out by reviewer #2, *PLA2G7*-deficient YCC16 cells are not very sensitive to ferroptosis compared with darapladib treatment, although the underlying mechanism is unclear. We suspect that *PLA2G7* WT and KO YCC16 cells generated using the CRISPR/Cas9 system may acquire overall ferroptosis resistance during single-cell cloning. We therefore re-established the *PLA2G7* WT/KO H1299 cell lines and found that the RSL3 sensitivity of the parental and cloned *PLA2G7* WT in H1299 cells was similar. Using these cells, we found that *PLA2G7* KO cells are hypersensitive to ferroptosis in response to RSL3 treatment or cysteine deficiency (revised Fig. 4a-j). As this reviewer suggested, we tested the effect of darapladib on ferroptotic cell death in WT and *PLA2G7* KO cells treated with various concentrations of RSL3 and found that darapladib no longer promoted ferroptosis in H1299/*PLA2G7* KO cells, suggesting that darapladib targets Lp-PLA2 (revised Fig. 4k).

Furthermore, we analysed the lipid profile of *PLA2G7* KO cells and found that most PE species were upregulated, whereas lysoPE was downregulated in a similar manner as that observed with darapladib treatment (revised Fig. 5f, g), suggesting that darapladib accelerates ferroptosis by increasing pro-ferroptotic phospholipids through Lp-PLA2 inhibition.

6. As in comment #5. In Fig 5K, please show the data using multiple doses of RSL3.

Response: We present new data using multiple doses of RSL3 in revised Fig. 4l, m. The re-expression of Lp-PLA2 in *PLA2G7* KO cells can partially restore ferroptosis resistance, possibly due to the limitation of transfection efficiency.

7. The authors only focused on the role of Lp-PLA2 and Dara in cancer cells. However, please examine whether Dara shows the ferroptosis sensitizing effect in non-cancer cell lines. The expression level of Lp-PLA2 in non-cancer cell lines also could be compared to that in cancer cells.

Response: This comment describes a very valuable point because early studies showed that Lp-PLA2 is mainly found in macrophages and endothelial cells. However, the roles of Lp-PLA2 in adipocytes, hepatocytes, and various cancer cells have recently been revealed, but very few studies have compared the expression of Lp-PLA2. During the revision, we tested the effect of darapladib in noncancer cells such as H9c2 cardiomyocytes and MEFs and found that darapladib also enhanced ferroptosis in these cells (revised Supplementary Figs. 3c, 4f). Unfortunately, there is no commercially available antibody that detects endogenous Lp-PLA2, which can be

validated in *PLA2G7* KO cells. In addition, because we used rat cardiomyocytes (H9c2), mouse MEFs, and human cancer cells, it was difficult to evaluate the differences in the expression of Lp-PLA2 among these cells.

8. *The synergistic effect of Dara with other classes of ferroptosis inducers other than GPX4 inhibitors should be checked. In the study, they used only GPX4 inhibitors and Cys depletion for induction of ferroptosis.*

Response: As suggested by the reviewer, we employed erastin, a well-established class I FIN, in H9c2 cells (revised Supplementary Fig. 4f). In addition, we added experiments using shGPX4 as suggested by the reviewer (revised Supplementary Fig. 4a-e).

9. *Please show the protective effect of Fer1 with a cell viability curve (Fig 1F, s2B) to show whether Fer-1 completely rescued the synergistic effect of RSL3 with Dara.*

Response: We confirmed that cotreatment with Fer-1 fully restored the synergistic effect of Dara and RSL3. We replaced the original data with new data (revised Fig. 1b and Supplementary Fig. 3b).

9. *In Fig 3D, it is well known that nutrients such as Vit E, selenium and CoQ10 contained in serum can suppress ferroptosis.*

Response: We thank the reviewer for this clear comment and have revised the sentence as follows:

“Interestingly, unlike lipoprotein deficiency, serum starvation did not alleviate ferroptosis, probably due to the concomitant depletion of anti-ferroptotic components in serum, such as vitamin E, selenium, and CoQ10, or a difference in composition between human and bovine serum.” (Page 8, lines 17-20)

10. *In Fig S4C, it is hard to compare the level of ACC. Please repeat the WB and quantify the band.*

Response: Because the phospho-ACC and total ACC bands in the original figure were quite faint, we repeated the WB analysis and present a clearer result with quantification of the pACC/ACC ratio (revised Supplementary Fig. 10d).

11. *In S5A, as Lp-PLA2 is not a gene name, siLp-PLA2 looks strange. Please show the efficiency of siPLA2G7 and PLA2G KO by WB if possible.*

Response: Thank you for pointing out our incorrect notation, and we have corrected the notation of gene names (revised Figs. 4, 5 and Supplementary Figs. 7, 8, 16). We have attempted to detect endogenous Lp-PLA2 protein using several commercially available antibodies (Antibodies online, ABIN653776 and LSBIO, LS-C295914), but we failed to specifically detect Lp-PLA2. Some of these antibodies detect a protein of approximately 75 kDa but also detect the same protein in *PLA2G7* KO cells. Instead, we tested the knockdown and knockout efficiency by RT-PCR and found that *PLA2G7* mRNA was barely detectable in the knockout cells but was detected at a level of approximately 10-20% in the knockdown cells.

References

1. Blackie JA, *et al.* The identification of clinical candidate SB-480848: a potent inhibitor of lipoprotein-associated phospholipase A2. *Bioorg Med Chem Lett* **13**, 1067-1070 (2003).
2. O'Donoghue ML, *et al.* Effect of darapladib on major coronary events after an acute coronary syndrome: the SOLID-TIMI 52 randomized clinical trial. *Jama* **312**, 1006-1015 (2014).
3. White HD, *et al.* Darapladib for preventing ischemic events in stable coronary heart disease. *N Engl J Med* **370**, 1702-1711 (2014).
4. Sun WY, *et al.* Phospholipase iPLA(2)beta averts ferroptosis by eliminating a redox lipid death signal. *Nat Chem Biol* **17**, 465-476 (2021).
5. Beharier O, *et al.* PLA2G6 guards placental trophoblasts against ferroptotic injury. *Proc Natl Acad Sci U S A* **117**, 27319-27328 (2020).
6. Sun WY, *et al.* Phospholipase iPLA2beta averts ferroptosis by eliminating a redox lipid death signal. *Nat Chem Biol* **17**, 465-476 (2021).
7. Lei G, Zhuang L, Gan B. mTORC1 and ferroptosis: Regulatory mechanisms and therapeutic potential. *Bioessays* **43**, e2100093 (2021).
8. Yi J, Zhu J, Wu J, Thompson CB, Jiang X. Oncogenic activation of PI3K-AKT-mTOR signaling suppresses ferroptosis via SREBP-mediated lipogenesis. *Proc Natl Acad Sci U S A* **117**, 31189-31197 (2020).
9. Zhang Y, *et al.* mTORC1 couples cyst(e)ine availability with GPX4 protein synthesis and ferroptosis regulation. *Nat Commun* **12**, 1589 (2021).
10. Chang MY, Han CY, Wight TN, Chait A. Antioxidants inhibit the ability of lysophosphatidylcholine to regulate proteoglycan synthesis. *Arterioscler Thromb Vasc Biol* **26**, 494-500 (2006).

11. Portman OW, Alexander M. Lysophosphatidylcholine concentrations and metabolism in aortic intima plus inner media: effect of nutritionally induced atherosclerosis. *J Lipid Res* **10**, 158-165 (1969).
12. Yamakawa T, *et al.* Lysophosphatidylcholine activates extracellular signal-regulated kinases 1/2 through reactive oxygen species in rat vascular smooth muscle cells. *Arterioscler Thromb Vasc Biol* **22**, 752-758 (2002).
13. Tyurin VA, *et al.* Oxidatively modified phosphatidylserines on the surface of apoptotic cells are essential phagocytic 'eat-me' signals: cleavage and inhibition of phagocytosis by Lp-PLA2. *Cell Death Differ* **21**, 825-835 (2014).
14. Yan B, *et al.* Membrane Damage during Ferroptosis Is Caused by Oxidation of Phospholipids Catalyzed by the Oxidoreductases POR and CYB5R1. *Molecular Cell* **81**, 355-369.e310 (2021).
15. Eaton JK, Furst L, Cai LL, Viswanathan VS, Schreiber SL. Structure-activity relationships of GPX4 inhibitor warheads. *Bioorg Med Chem Lett* **30**, 127538 (2020).
16. Wiernicki B, *et al.* Excessive phospholipid peroxidation distinguishes ferroptosis from other cell death modes including pyroptosis. *Cell Death Dis* **11**, 922 (2020).
17. Kim OH, *et al.* Externalized phosphatidylinositides on apoptotic cells are eat-me signals recognized by CD14. *Cell Death Differ* **29**, 1423-1432 (2022).

REVIEWERS' COMMENTS

Reviewer #1 (Remarks to the Author):

In the revised version of the manuscript, authors addressed most of the points from the 1st round of revision and supplements the revised version of the manuscript with significant amount of new experimental data. Although, some of the comments remained unanswered.

Thus for instance to the comment

"Figure 3E – GPX4 does look decreased to me on the blots upon darapladib treatment... please provide Source Data/ raw image files. Authors state "we observe no significant differences in protein expression" which does not seem to be right for a number of proteins show on the immunoblot Figure 3E – ELOVL5, LPCAT3 (both decreasing), FSP1 (increasing) and GPX4. Please provide raw data in replicates for reanalysis."

Author response in the revised version of the manuscript:

"we observed no significant differences in protein expression upon RSL3 and/or darapladib treatment" (Page 8, lines 27-28).

As well as the following comment was not addressed.

"Lipidomics experiments should be performed with darapladib in combination with RSL3 treatment as well so judge the role of the Lp PLA2 inhibition on ferroptosis. Comparison of DMSO vs darapladib doesn't tell us anything about the ferroptosis related remodelling of the lipidome. According to the data shown on Figure S1A 2 uM darapladib does not alter cell viability. Thus I am wondering how these lipidomics results can be connected with ferroptosis. Presented results rather reflect lipidome alterations in the presence of PLA2 inhibition and thus might indicate preferential substrates for the enzyme. Please present results accordingly."

Lipidomics data are provided as e.g. PC 40:4(18:0/20:4) – it is rather unconventional notation and I doubt that the sn-1 vs sn-2 positions were resolved. So please report as PC 18:0_20:4.

Overall, due to the complexity of the study, and improved accuracy of the revised manuscript, I would recommend it to be accepted for the publication after lipid annotation are be corrected.

Reviewer #2 (Remarks to the Author):

The authors have provided additional data to support their conclusion. The current version of the manuscript has substantially improved and most of my remarks have been addressed.

Reviewer #3 (Remarks to the Author):

The authors have appropriately addressed the comments raised by the reviewer by presenting additional data, which would suffice to conclude the study. Notably, the data in Fig4K robustly demonstrate the on-target effect of Dara for ferroptosis-sensitization via PLA2AG-dependending manner.

However, the reviewer has a minor comment regarding the in vivo effect of PACMA31. Although the GPX4 inhibiting activity of PACMA31 in cultured cells is evident, it has not been robustly confirmed whether the synergistic antitumor effect of PACMA31 is due to its ferroptosis-inducing activity, as supported only by the rescuing effect of Fer-1 treatment. The lack of a Fer-1 alone group (without PACMA-31 and Dara) means that it is unclear whether the effect of Fer-1 is to rescue the effect of PACMA-31+Dara or simply promote tumor growth by Fer1 itself. Additionally, the lack of (oxi)lipidomics data from tumor samples is a limitation of the model for demonstrating whether PACMA31 actually induces ferroptosis in the in vivo setting. While the reviewer does not request further investigation into this issue given that the ferroptosis-sensitizing effect of Dara has been thoroughly examined in other data sets, it remains a topic for future research beyond the present study. Therefore, the reviewer suggests that the claim that the model induced by PACMA31 is ferroptosis in vivo be weakened and that limitation be additionally stated to avoid inappropriate use of

this drug as an in vivo available ferroptosis inducer/sensitizer in other future studies without further validation.

Author Rebuttals to First Revision:

Reviewer #1 (Remarks to the Author):

In the revised version of the manuscript, authors addressed most of the points from the 1st round of revision and supplements the revised version of the manuscript with significant amount of new experimental data. Although, some of the comments remained unanswered.

Response: We would like to thank to the reviewers for their time and constructive comments on our manuscript

Thus for instance to the comment

“Figure 3E – GPX4 does look decreased to me on the blots upon darapladib treatment... please provide Source Data/ raw image files. Authors state “we observe no significant differences in protein expression” which does not seem to be right for a number of proteins show on the immunoblot Figure 3E – ELOVL5, LPCAT3 (both decreasing), FSP1 (increasing) and GPX4. Please provide raw data in replicates for reanalysis.”

Author response in the revised version of the manuscript:

“we observed no significant differences in protein expression upon RSL3 and/or darapladib treatment” (Page 8, lines 27-28).

Response: While our repeated experiments show no significant difference in darapladib-treated cells, we observed that the levels of several proteins are slightly changed in PLA2G7-KO cells, but the changes were not consistent between YCC-16 and H1299 cells. Therefore, we added the following sentences as follows:

“Similar to the findings with darapladib, PLA2G7-deleted cells showed no significant alterations in several key ferroptosis regulators although some fluctuations in the protein levels were observed (Supplementary Fig. 8i).” (Page 10, lines 2-4). We also have provided raw data in source data file.

As well as the following comment was not addressed.

“Lipidomics experiments should be performed with darapladib in combination with RSL3 treatment as well so judge the role of the Lp PLA2 inhibition on ferroptosis. Comparison of DMSO vs darapladib doesn’t tell us anything about the ferroptosis related remodelling of the lipidome. According to the data shown on Figure S1A 2 uM darapladib does not alter cell viability. Thus I am wondering how these lipidomics results can be connected with ferroptosis. Presented results rather reflect lipidome alterations in the presence of PLA2 inhibition and thus might indicate preferential substrates for the enzyme. Please present results accordingly.”

Response: Although we directly analysed lipidomic changes upon simultaneous treatment with darapladib and RSL3, we suggest that darapladib can rapidly alter specific phospholipids abundance such as PE-38:4, resulting in the sensitization to ferroptosis. Since RSL3 can also affect lipidome via oxidative degradation, most studies investigating lipidomic under ferroptotic condition show lipidomic alteration without ferroptosis induction. For example, recent Cell paper by Xuejun Jiang group also shows lipidomic changes under MBOAT1 overexpression or knockout without ferroptosis inducers¹. Nevertheless, as concerned by this reviewer, we included the limitation of our study as follows:

Ferroptosis surveillance independent of GPX4 and differentially regulated by sex hormones

“These data suggest that inhibition of Lp-PLA2 leads to the accumulation of AA and AdA containing PE or PE-p, rendering cells sensitive to ferroptosis although lipidomic changes under simultaneous inhibition of Lp-PLA2 and GPX4 were not directed determined” (Page 11, lines 16-18)

Lipidomics data are provided as e.g. PC 40:4(18:0/20:4) – it is rather unconventional notation and I doubt that the sn-1 vs sn-2 positions were resolved. So please report as PC 18:0_20:4.

Overall, due to the complexity of the study, and improved accuracy of the revised manuscript, I would recommend it to be accepted for the publication after lipid annotation are be corrected.

Response: Thank you for your suggestion. We have revised the lipid annotation.

Reviewer #2 (Remarks to the Author):

The authors have provided additional data to support their conclusion. The current version of the manuscript has substantially improved and most of my remarks have been addressed.

Response: We greatly appreciate the positive evaluation of our research.

Reviewer #3 (Remarks to the Author):

The authors have appropriately addressed the comments raised by the reviewer by presenting additional data, which would suffice to conclude the study. Notably, the data in Fig4K robustly demonstrate the on-target effect of Dara for ferroptosis-sensitization via PLA2AG-dependending manner.

Response: We appreciate the thorough review and positive feedback of our work.

However, the reviewer has a minor comment regarding the in vivo effect of PACMA31. Although the GPX4 inhibiting activity of PACMA31 in cultured cells is evident, it has not been robustly confirmed whether the synergistic antitumor effect of PACMA31 is due to its ferroptosis-inducing activity, as supported only by the rescuing effect of Fer-1 treatment. The lack of a Fer-1 alone group (without PACMA-31 and Dara) means that it is unclear whether the effect of Fer-1 is to rescue the effect of PACMA-31+Dara or simply promote tumor growth by Fer1 itself. Additionally, the lack of (oxi)lipidomics data from tumor samples is a limitation of the model for demonstrating whether PACMA31 actually induces ferroptosis in the in vivo setting. While the reviewer does not request further investigation into this issue given that the ferroptosis-sensitizing effect of Dara has been thoroughly examined in other data sets, it remains a topic for future research beyond the present study. Therefore,

the reviewer suggests that the claim that the model induced by PACMA31 is ferroptosis in vivo be weakened and that limitation be additionally stated to avoid inappropriate use of this drug as an in vivo available ferroptosis inducer/sensitizer in other future studies without further validation.

Response: Thanks again for this suggestion. As PACMA31 showed only minimal efficacy for tumor suppression although it synergizes with darapladib in vivo which can be rescued by ferrostatin-1, we added the following sentences as follows:

“However, due to the low in vivo potency and the lack of oxidized lipid analysis, whether PACMA31 is a reliable ferroptosis inducer in mice requires further investigation.” (Page 17, line 18-20)

References

1. Liang D, *et al.* Ferroptosis surveillance independent of GPX4 and differentially regulated by sex hormones. *Cell* **186**, 2748-2764 e2722 (2023).